# Repurposing clinically safe drugs for DNA repair pathway choice in CRISPR genome editing and synthetic lethality

Dominik Macak [1,2], Philipp Kanis [1,2] & Stephan Riesenberg [1]✉

We evaluate the effect of most FDA-approved drugs (>7,000 conditions) on double-strand DNA break repair pathways by analyzing mutational outcomes in human induced pluripotent stem cells. We identify drugs that can be repurposed as inhibitors and enhancers of repair outcomes attributed to non-homologous and microhomology-mediated end joining (NHEJ, MMEJ), and homology-directed repair (HDR). We also identify functions of the proteins estrogen receptor 2 (ESR2) and aldehyde oxidase 1 (AOX1), affecting several key DNA repair proteins, such as ATM and 53BP1. Silencing of ESR2 can have a synergistic effect on increasing HDR when combined with NHEJ inhibition (mean 4.6-fold increase). We further identify drugs that induce synthetic lethality when NHEJ or HDR is blocked and may therefore be candidates for precision medicine. We anticipate that the ability to modulate the DNA repair outcomes with clinically safe drugs will help disease modeling, gene therapy, chimeric antigen receptor immunotherapy, and cancer treatment.

DNA double-strand breaks (DSBs) occur spontaneously at random genomic sites due to metabolically generated reactive oxygen species, and higher levels of DSBs can be induced by exogenous agents, such as ionizing radiation and DNA damaging chemicals used in cancer chemotherapy[1]. In contrast, CRISPR nucleases complexed with a guide RNA (gRNA) can introduce DSBs at predetermined genomic sites[2].

Cellular DSBs are primarily repaired through non-homologous end joining (NHEJ) while microhomology-mediated end joining (MMEJ) acts as a backup pathway[3] (Fig. 1a). Both NHEJ and MMEJ typically produce insertions and deletions (indels) of a few nucleotides at the DSB, but they can also cause larger deletions spanning several hundred nucleotides or chromosomal rearrangements[4,5]. This is widely used for targeted disruption of genes after CRISPR-induced DSBs in disease modeling[6], genome-wide knock-out screens[7], chimeric antigen receptor (CAR-T) cell enhancement for immuno-oncology[8], as well as in gene therapy[9] (Supplementary Fig. 1). Notably, the first CRISPR gene therapy was approved by regulatory authorities in the US, UK, and EU in late 2023[10]. The composition of DNA sequences adjacent to DSBs can result in preferential repair by either NHEJ, forming mainly insertions of one base pair (bp), or by MMEJ, forming predictable deletions;

both pathways can be used for somewhat predictable and template-free therapeutic gene editing[11,12]. NHEJ and MMEJ can also be used to integrate genes encoded by double-stranded DNA donors by homology-independent knock-in[13] and microhomology-dependent knock-in[14–16], respectively.

In addition, DSBs can be repaired through less efficient homologous recombination (HR) using sister chromatids as templates and related homology-directed repair (HDR), in which exogenous DNA donors are used as templates. When a single-stranded DNA donor is used, HDR is sometimes referred to as single-strand template repair (SSTR)[17–19]. This precise repair can introduce all 12 types of point mutations, insertions, and deletions, as well as entire genes into the genome. Template-directed precise genome editing holds great promise for the treatment of genetic diseases, where disease-causing alleles cannot simply be removed, but must be restored to their healthy wild type states[20,21]. It is also used for knock-in of CARs in T cells engineered to treat hematological malignancies[8].

If none of the repair pathways succeeds to repair the DSB, cells will die by DNA damage-induced apoptosis[22]. Acquired or inherited defects in repair genes can create specific vulnerabilities in cancer cells,

[1]Max Planck Institute for Evolutionary Anthropology, Leipzig, Germany. [2]These authors contributed equally: Dominik Macak, Philipp Kanis.
✉e-mail: stephan_riesenberg@eva.mpg.de

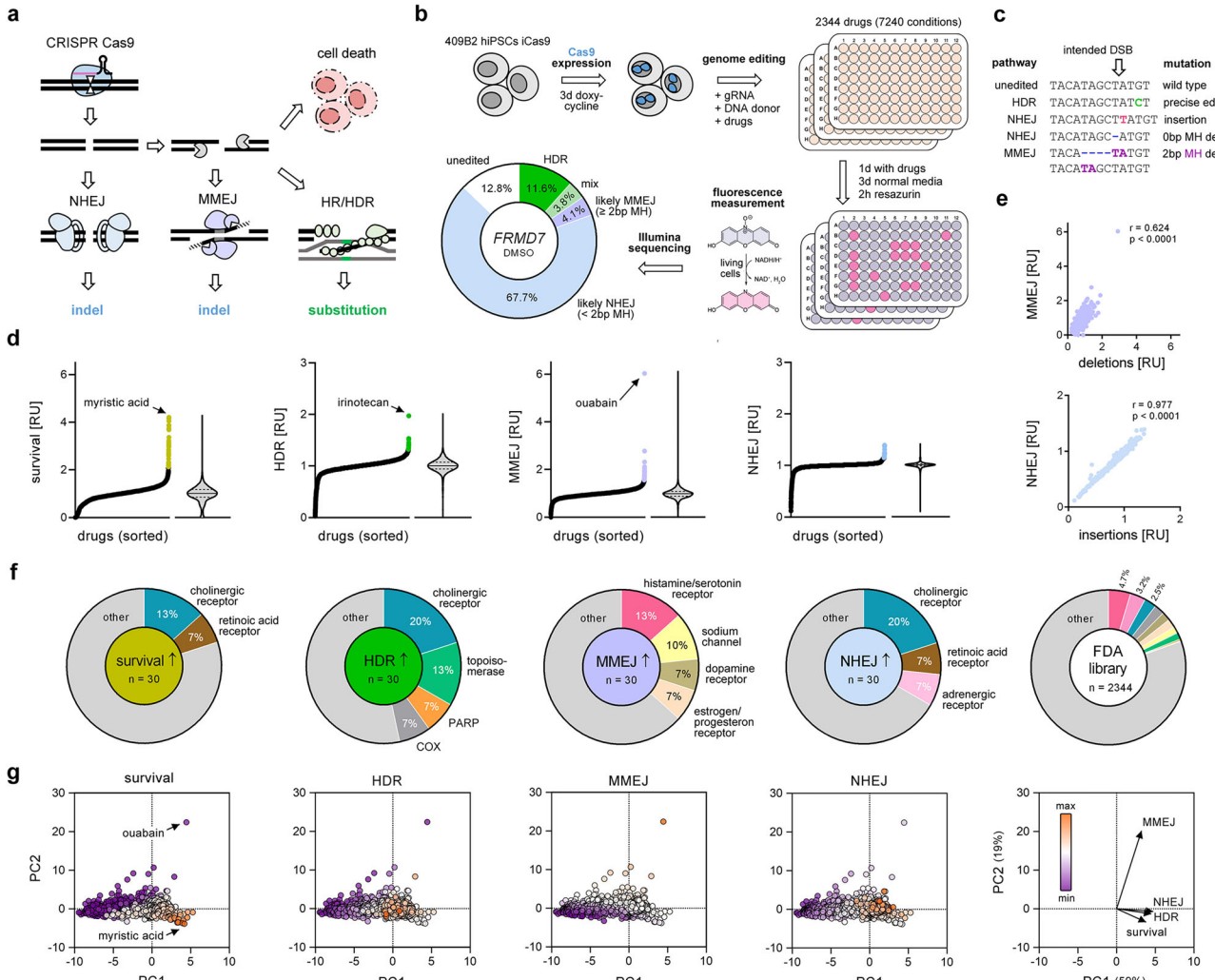

**Fig. 1 | Drug screening for DSB repair pathway modulators. a** Schematic of cellular DNA double-strand break (DSB) repair pathways and quantifiable outcomes of CRISPR genome editing. **b** Drug repurposing screen workflow. 409B2 hiPSCs expressing a doxycycline-inducible Cas9 (iCRISPR) are used to edit a genomic target (*FRMD7*) during drug treatment. After recovery in normal media cell survival is measured using a resazurin fluorescence assay, followed by DNA extraction and Illumina sequencing. The pie chart shows quantified editing outcomes of mock treatment with DMSO. One replicate for each screening condition was performed (n = 7240). **c** Schematic of the sequencing results after CRISPR editing and their assignment to a particular DNA repair pathway. **d** The effect on drug conditions on the distribution of changes in cell survival (light gold), precise edits (HDR; green), indels with <2 bp microhomology (NHEJ; light blue), and deletions ≥2 bp microhomology (MMEJ; light purple). Shown are relative values compared to the respective mock treatments. The top 30 drugs in each distribution are highlighted in color. **e** Correlation between deletions and insertions with MMEJ and NHEJ, respectively. Pearson correlation (r) and *p* value are stated (two-tailed, no adjustments made for multiple comparisons). All *p* values are <0.0001. **f** Target class enrichment of top 30 colored drugs from **d** for survival, HDR, MMEJ, and NHEJ. Depicted are target classes that are targeted by at least two different drugs within the top 30 drugs. For comparison the share of the same target classes within the whole FDA library is shown (right pie chart). **g** Principal component analysis (PCA) of obtained metrics to reduce dimensionality. Shown is the separation of the data by PC1 and PC2 and circles are colored corresponding to their relative values in cell survival, HDR, MMEJ, and NHEJ, respectively. The right panel shows the direction of the PCA vectors per metric and percentage of data variation explained by PC1 and PC2. Source data are provided as a Source data file.

providing an opportunity for targeted therapy by synthetic lethality, where pharmacological inhibition of a compensatory pathway is toxic to cancer but not to healthy cells[23].

Several different small molecules have been used to alter DSB repair pathway choice[24,25]. HDR can be reduced by inhibiting the central HR protein RAD51 with B02[21,26], NHEJ can be blocked by inhibiting DNA-PKcs with e.g. M3814 or AZD7648[27,28], and MMEJ can be blocked by inhibiting Polθ, the protein encoded by *POLQ*, with ART558[29] or by inhibiting PARP1 with rucaparib[12]. PARP inhibitors are best known for inducing synthetic lethality in *BRCA1/2* negative, and thus HR-deficient, tumors[30,31]. Almost exclusive HDR can be achieved by combined inhibition of NHEJ and MMEJ[21,32]. HDR can also be increased by cell cycle arrest to enrich cells in phases where HR factors are present[33], as well as by activation of RAD51[34,35], but the latter increase could not be replicated in several studies[26,36,37]. To date, no small molecule enhancers of NHEJ or MMEJ have been described. Currently, only PARP inhibitors have US Food and Drug Administration (FDA) approval for treatment of various cancers via synthetic lethality[38] while M3814, AZD7648, and ART558 are in clinical trials for the treatment of cancer[23,39–41]. If proven safe, they would be promising candidates for clinical applications of genome editing and oncology.

Here, we elucidate the effects of essentially all small molecule drugs that had FDA-approval by 2023[42,43] on DSB repair pathway choice and cell survival during CRISPR editing in human induced pluripotent stem cells (hiPSCs). We identify clinically safe inhibitors and enhancers of mutational outcomes attributed to NHEJ, MMEJ, and HDR, as well as novel protein targets that modulate DNA repair. This expands the

toolbox for DNA repair modulation and could potentially be used in clinical applications (Supplementary Fig. 1).

## Results

### Drug screening for DSB repair modulators

We electroporated chemically synthesized gRNA (crRNA/tracrRNA hybrid) targeting a site in *FRMD7* and a single-stranded DNA donor carrying a point mutation into hiPSCs that expressed Cas9 after induction by doxycycline (iCRISPR)[26] (Fig. 1b). After electroporation, the cells were seeded into 96-well plates and exposed to drugs (10 μM, 1 μM, and 0.1 μM) for 24 h. Three days after editing, we assessed cell viability using a resazurin assay[44], and isolated DNA for targeted PCR amplification and subsequent Illumina sequencing. For 73 drugs that showed high toxicity at the lowest tested concentration (0.1 μM; relative viability <20%), we additionally tested two lower concentrations (0.01 μM, 0.001 μM), resulting in a total of 7,240 tested conditions of different drug concentrations (2,344 total drugs). We have previously shown that the *FRMD7* site can be efficiently edited and that big deletions, which would not be quantified by target PCR amplification, are very rare for this target (2% of single cell-derived clones)[27]. We scored sequences with the intended precise nucleotide substitution as derived from HDR. Deletions were classified as MMEJ if one end of the deleted sequence matched the undeleted sequence at the other end by two or more nucleotides (≥2 bp microhomology) (Fig. 1c). Insertions and deletions with <2 bp microhomology were attributed to NHEJ as reported previously[27]. In addition, we report the share of insertions and deletions for all indels, as well as 'mix' to describe combinations of targeted nucleotide substitutions and indels (FDA drug screen result catalog: Supplementary Data 1).

The effect of drug conditions on the range and distribution of changes in cell survival, precise edits (HDR), indels with <2 bp microhomology (NHEJ), and deletions ≥2 bp microhomology (MMEJ) is shown in Fig. 1d. Deletions or insertions were highly correlated with MMEJ or NHEJ, respectively (Fig. 1e). Analysis of protein targets from the top 30 drug hits that increased cell survival or mutational outcomes attributed to either HDR, NHEJ, or MMEJ, showed higher enrichment of certain target classes compared to their proportion in the drug library (Fig. 1f). Top drugs that increased cell survival, HDR, and NHEJ were highly enriched in targets affecting cholinergic receptors (13–20% of top hits, random expectation: 2.5%). We found a potent HDR increase with topoisomerase inhibitor irinotecan (2-fold, 1 μM), albeit associated with 90% cell death. Ouabain, which inhibits the sodium-potassium pump, resulted in comparable cell death, but strikingly increased MMEJ (6-fold, 10 μM). Ouabain was the strongest outlier when using principal component analysis (PCA) to reduce the dimensionality of our data, and PC1 and PC2 were largely driven by MMEJ and cell survival, respectively (Fig. 1g). We observe that high cell death was associated with reduced genome editing efficiencies. More than 10% of all drugs killed at least 95% of cells at a concentration of 10 μM, and on average, a third of the top 30 hits that enhanced tested repair outcomes were drugs at 10 μM (Supplementary Fig. 2a,b). Since this typical high-throughput screening concentration[45] would result in missing potential hits for other FDA drug screens, and lowering the concentration to 1 μM would reduce toxicity-related dropout at the cost of reduced power to detect biological effects, we utilized our information on the concentration-dependent toxicity of all drugs to provide a single toxicity-reduced screening concentration for each drug for future FDA drug repurposing screens (Supplementary Data 2).

### Repair pathway outcome modulator drugs

To find drugs that affect specific repair pathways, we used quartile-dependent filters to classify compounds as modulators (inhibitors or enhancers) of outcomes attributed to HDR, MMEJ, and NHEJ,

respectively (Supplementary Fig. 2c) (see "Methods"). For example, a drug that reduces both targeted substitutions (HDR down) and indels (NHEJ and/or MMEJ down) cannot be considered an HDR pathway inhibitor; an HDR pathway inhibitor is expected to reduce targeted substitutions while leaving indels unchanged or even increasing indels. We recognize that this approach focuses on the outcome of DNA repair, but cannot distinguish between direct effects of drugs on DNA repair proteins and indirect secondary epiphenomena. Categorization of drugs resulted in clusters in the PCA for MMEJ modulators (PC1 and 2), as well as HDR and NHEJ modulators (PC3 and 4) (Fig. 2a).

Due to the large number of potential modulators identified, we selected a subset of 55 drugs to perform dose-response relationships for iCRISPR-Cas9 editing of *FRMD7* (see "Methods") (Fig. 2b). Among these, 22 drugs increased or decreased a tested metric by at least 20% (Fig. 2b), while 23 drugs had a smaller or no effect, and 10 drugs decreased all tested metrics with increasing concentration-dependent toxicity (Supplementary Data 3). Heatmaps of the effect of the 22 drugs that can be repurposed to affect outcomes attributed to HDR, NHEJ, MMEJ and/or survival at effective concentrations are shown in Fig. 2c, with corresponding dose-response curves shown in Supplementary Fig. 3. In addition to the strongest modulator ouabain, we selected modulators with low or moderate toxicity for editing a site in *NOVA1* and obtained comparable effects on DNA repair outcomes for *FRMD7/NOVA1*: ouabain 307%/200% MMEJ increase; duloxetine 187/153% MMEJ increase; artemether 75/85% MMEJ decrease; cyproterone 115/119% HDR increase; cytarabine 29/45% HDR decrease, myristic acid 149/120% survival increase (Fig. 2d). Both ouabain and duloxetine resulted in a clear change in the deletion shape patterns (Supplementary Fig. 4). Testing selected modulators in HEK293 and K562 cells showed comparable tendencies for repair outcomes, albeit with sometimes weaker effects (Supplementary Fig. 5). This, and the fact that the tested ouabain concentration was lethal to HEK293 and K562 cells, indicates that drug concentrations may need to be optimized for different cell types.

For editing of *FRMD7* in HEK293 and K562 cells, as well as for *NOVA1* in hiPSCs, cytarabine not only decreased HDR and MMEJ, but also increased NHEJ by 119–131%. We additionally tested the effect of cytarabine on the *VCAN* target in iCRISPR-Cas9 hiPSCs, as well as *SV2C* in HEK293 cells, and *FRMD7* in K562 and THP1 cells. Cytarabine decreased HDR for all targets by around a half and also increased NHEJ (Supplementary Fig. 6a). In addition to inferring DNA repair outcomes by targeted sequencing analysis, we also tested these using extra-chromosomal DNA DSB repair reporter assays for HR, MMEJ, and NHEJ, respectively[46]. In these assays, successful repair of broken DNA constructs results in functional luciferase and emission of bioluminescence. Cytarabine treatment reduced bioluminescence for HR, MMEJ, and NHEJ reporters, indicating inhibition of all those repair pathways (Supplementary Fig. 6b).

Because short-term drug toxicity is tolerable for cell line engineering in which surviving cells are propagated, we also tested combinations of drugs, independent of their toxicity, to explore potential additive effects for both *FRMD7* and *NOVA1* in modulating HDR outcomes. A combination of olaparib with isosorbide mononitrate and hodostin increased HDR 1.7-fold for both genes, and cytarabine and dronedarone showed an additive effect in reducing HDR (Supplementary Fig. 7). In addition, we investigated the effect of repurposed drugs on outcomes of prime editing, which should not depend on canonical DSB repair pathways[47]. Surprisingly, dronedarone can slightly increase prime editing efficiency of both *FANCF* and *RNF2* (1.1- and 1.3-fold) (Supplementary Fig. 8).

In our initial screen, drugs affecting cholinergic receptors accounted for 13–20% of the top hits that increased survival, HDR, and NHEJ, respectively (Fig. 1f). Among anticholinergic drugs, tolterodine and orphenadrine increased the combined total editing the most (115% increase) (Supplementary Data 1). Therefore, we further investigated

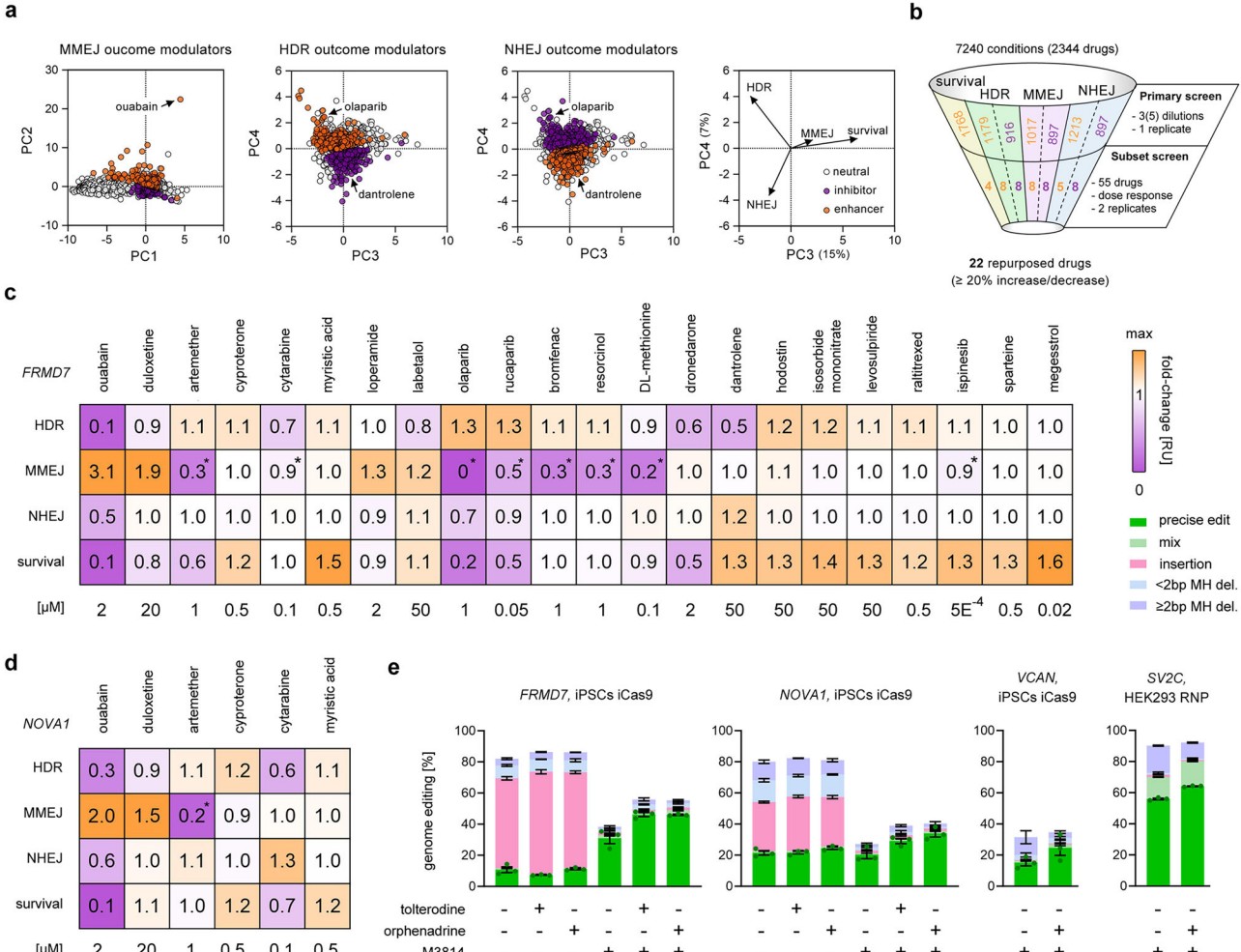

**Fig. 2 | Repurposed DSB repair modulator drugs. a** Principal component analysis (PCA) of the initial drug screen with circles colored based on categorization using metric quartiles into enhancers, inhibitors, and neutral compounds for outcomes attributed to MMEJ, HDR, and NHEJ. PC1 and PC2 separate MMEJ, whereas. PC3 and PC4 separate HDR and NHEJ. The right panel shows the direction of the PCA vectors per metric and percentage of data variation explained by PC3 and PC4. For categorization filters, see Supplementary Fig. 2c. **b** Numbers of categorized enhancers and inhibitors from the initial screen, as well as those of the subset screen. Enhancers shown in orange, inhibitors in purple. **c** Heatmap of relative fold-changes of HDR, MMEJ, NHEJ, and survival using an inducible Cas9 in 409B2 hiPSCs to edit *FRMD7* using different drugs and their respective effective concentration. Asterisks

indicate normalized MMEJ changes based on the method's detection limit for MMEJ reduction (full inhibition = 0.74 for *POLQ* knock-out[21]). **d** Heatmap of selected drugs at effective concentration with relative fold-changes of HDR, MMEJ, NHEJ, and survival using an inducible Cas9 in 409B2 hiPSCs to edit *NOVA1*. **e** Effects of combinations of tolterodine (10 μM), orphenadrine (10 μM), and M3814 (2 μM) on genome editing efficiencies of *FRMD7*, *NOVA1*, and *VCAN* in iCas9 409B2 hiPSCs and of *SV2C* with Cas9-HiFi RNP in HEK293. For precise edits, replicates are depicted by dots. Independent biological replicates were performed (n = 2 for **b**, **c**; n = 6 for **d** ouabain, duloxetine, cyproterone, n = 3 for **d** artemether, cytarabine, myristic acid; n = 3 for **e**). Error bars indicate the s.e.m. Source data are provided as a Source Data file.

the effect of both drugs with or without inhibition of NHEJ by the DNA-PKcs inhibitor M3814. For both *FRMD7* and *NOVA1*, M3814 drastically reduced indels so that 81% and 75% of all editing events different from the wild type were due to HDR (Fig. 2e). Addition of either tolterodine or orphenadrine increased HDR 1.4- to 1.7-fold compared to M3814 alone. A synergistic effect was also observed for editing of *VCAN*, a target with common residual MMEJ-derived deletions, as well as for *SV2C* in HEK293 cells. To gauge a mechanistic insight on how orphenadrine can modulate DNA repair, we treated hiPSCs with or without orphenadrine, M3814, and a combination of both, and stained for γ-H2AX (S139) and 53BP1. γ-H2AX is an early marker of DSBs, and 53BP1 is recruited to DSB sites and involved in DSB pathway choice between NHEJ and HDR. Interestingly, orphenadrine and the combination of orphenadrine with M3814 resulted in 3.2-fold and 2.2-fold more 53BP1 foci per nucleus, while M3814 alone did not increase foci (Supplementary Fig. 9). No change of rare γ-H2AX staining was observed in any condition.

**Novel protein targets that modulate DSB repair outcomes**

In addition to cholinergic receptors, our initial screen identified several other protein classes as targets of drugs that enhanced outcomes attributed to HDR, NHEJ, and MMEJ, respectively (Fig. 1f). We thus proceeded to make single-cell-derived knock-out cell lines for enriched protein classes using CRISPR editing of 409B2 iCas9 hiPSCs. Since protein classes are comprised of several genes, we chose to target the gene with the highest expression in our 409B2 hiPSC line[48]. We generated biallelic knock-outs of seven genes, could only obtain a monoallelic knock-out clone of poly(ADP-ribose)polymerase 1 (*PARP1*), and were unable to generate a knock-out clone of the gene DNA topoisomerase II alpha (*TOP2A*), likely due to gene essentiality[49] (Supplementary Fig. 10).

Editing of *FRMD7* in a cell line with a knock-out of estrogen receptor 2 (*ESR2*) resulted in a 1.4-fold increase in HDR and a 1.7-fold increase in cell survival compared to 409B2 iCas9 hiPSCs with wild type *ESR2* (Fig. 3a). For targeting *NOVA1*, the knock-out of *ESR2* alone

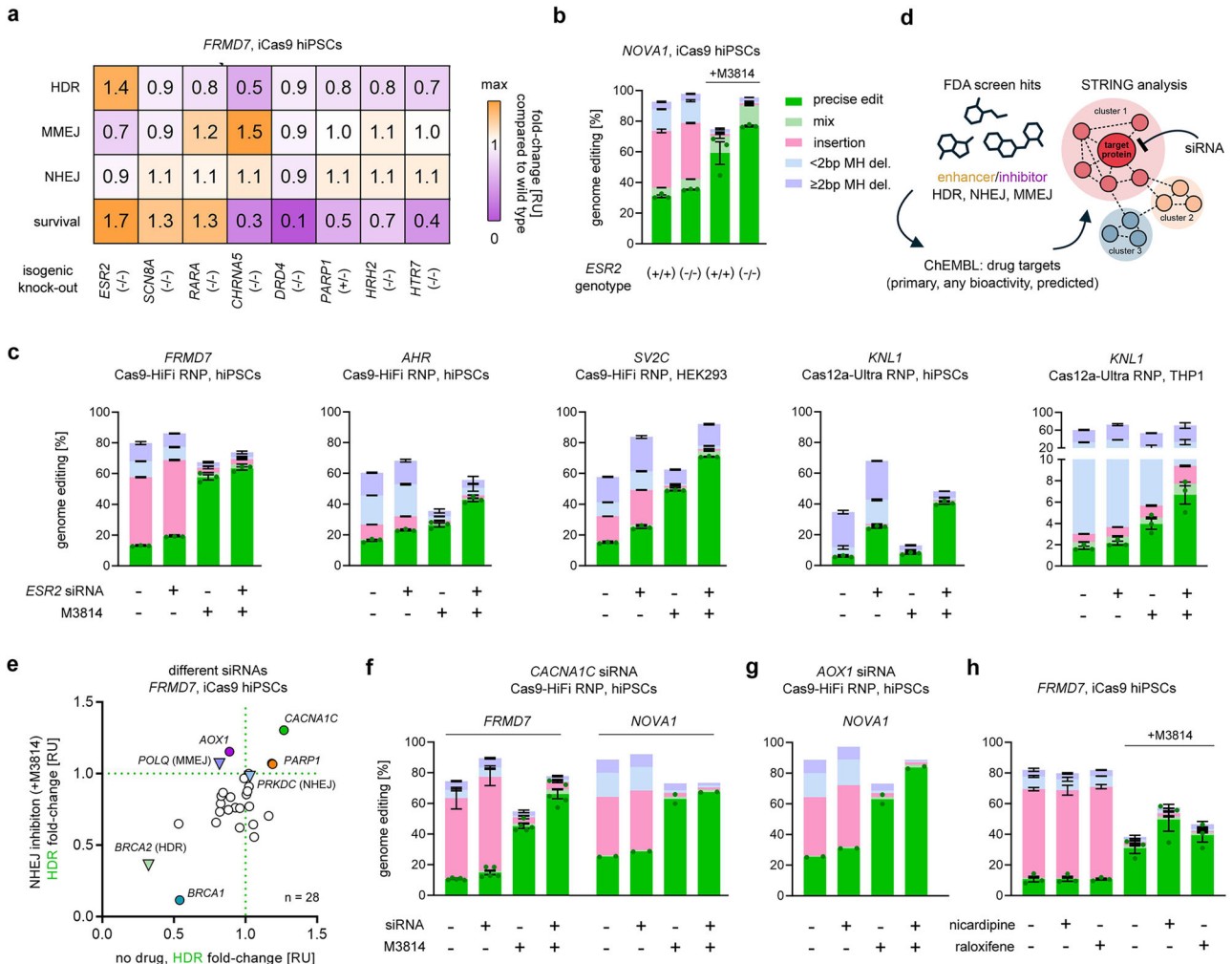

**Fig. 3 | Novel protein modulators of DSB repair pathways. a** Heatmap of relative fold-changes of HDR, MMEJ, NHEJ, and survival using an inducible Cas9 in 409B2 hiPSCs to edit *FRMD7* in isogenic single cell-derived knock-out cell lines. **b** Genome editing efficiencies of *NOVA1* in wild type and *ESR2* knock-out iCas9 hiPSCs, with or without M3814. For precise edits, replicates are depicted by dots. **c** Effects of siRNA-mediated silencing of *ESR2* on genome editing efficiencies for *FRMD7* and *AHR* with Cas9-HiFi RNP in 409B2 hiPSCs, *SV2C* with Cas9-HiFi RNP in HEK293, as well as *KNL1* with Cas12a-Ultra RNP in HEK293 and THP1, with or without M3814. **d** Schematic of novel DSB repair protein identification. The top 30 hits from the initial drug screen, categorized as enhancers or inhibitors of outcomes attributed to HDR, NHEJ, and MMEJ were used as input to retrieve protein targets from the ChEMBL database. Protein targets were subjected to STRING analysis to identify protein-protein interaction networks containing potential targets for siRNA-mediated knock-down of the corresponding mRNA during CRISPR editing. **e** Effects of lipofection of

siRNAs targeting mRNAs of 25 STRING-predicted genes or control siRNAs (*PRKDC*, *BRCA2*, *POLQ*) on relative fold-change of HDR for editing *FRMD7* in 409B2 iCRISPR Cas9 hiPSCs. The *X*-axis shows effects when no drug is added, and the *Y*-axis shows effects compared to when NHEJ is inhibited by 2 μM M3814. MMEJ inhibitors preferably increase HDR when NHEJ is already inhibited. **f** Effects of siRNA-mediated silencing of *CACNA1C* on genome editing efficiencies of *FRMD7* and *NOVA1* with Cas9-HiFi RNP in 409B2 hiPSCs, with or without M3814. **g** Effects of siRNA-mediated silencing of *AOX1* on genome editing efficiencies of *NOVA1* with Cas9-HiFi RNP in 409B2 hiPSCs, with or without M3814. **h** Effects of nicardipine or raloxifene on genome editing efficiencies of *FRMD7* with Cas9-HiFi RNP in 409B2 hiPSCs, with or without M3814. Independent biological replicates were performed (*n* = 3 for **a**–**c**, **e**, **h**, *n* = 5 *FRMD7* in **f**, **j**; *n* = 2 for *NOVA1* in **f** and **g**). Error bars indicate the s.e.m. Source data are provided as a Source data file.

also increased HDR and demonstrated an additive effect with M3814 (Fig. 3b). We confirmed the increased HDR capabilities of *ESR2* knock-out cells by a 1.5-fold increased bioluminescence of the extrachromosomal HR repair reporter assay (Supplementary Fig. 11). Also, NHEJ and MMEJ reporter signals were increased by 1.5- and 2.1-fold, respectively. We continued to test a transient approach by delivery of HiFi-Cas9 RNP[50] and DNA donor together with siRNA targeting the mRNA coding for *ESR2* (Fig. 3c). This transient inhibition achieved a similar level of HDR increase as observed in the *ESR2* knock-out cell line for the *FRMD7* target and had an additive effect when combined with M3814. When editing *AHR* with Cas9-HiFi in the same cell line, *KNL1* with Cas12a-Ultra[51], and *SV2C* with Cas9-HiFi in HEK293 cells, we even observed a synergistic effect of *ESR2* siRNA and M3814 to increase HDR efficiency (*ESR2* siRNA: mean 2.5-fold, M3814: mean 2.1-fold, combined:

mean 4.8-fold). For editing of *KNL1* with Cas12a-Ultra in THP1 cells, which have a very low inherent HDR efficiency, we achieved a synergistic increase from 1.7 to 6.7%.

Motivated by the discovery of the relevance of ESR2 in genome editing using knock-out cell lines, we continued with siRNA inhibition of additional candidate genes, as this allows for higher throughput to find more protein targets that modulate DNA repair outcomes. To do this, we selected the top 30 drugs from our initial screen that enhanced or inhibited HDR, NHEJ, and MMEJ, respectively, retrieved their primary targets and targets with published associated bioactivity data from the ChEMBL database[52], as well as predicted targets[53] (see "Methods"), yielding lists of proteins associated with each drug. We performed functional enrichment analysis and used the STRING web tool[54] to identify protein-protein interaction networks (Fig. 3d and

Supplementary Data 4). For each protein interaction cluster, we selected the protein with the highest number of interactions within the cluster for siRNA-mediated knock-down of the corresponding mRNA during CRISPR editing. In total, we selected siRNAs targeting 25 STRING-predicted modulators (30 clusters, five non-unique shared modulators), as well as control siRNAs against *BRCA2* (HDR inhibition), *PRKDC* (NHEJ inhibition), and *POLQ* (MMEJ inhibition). To this end, we lipofected siRNAs during editing of *FRMD7* in iCRISPR Cas9 cells with or without M3814 to also detect additive effects with NHEJ inhibition (Fig. 3e). For completeness, we also performed editing with simultaneous small molecule-induced inhibition of MMEJ by ART558 or HDR by B02 (Supplementary Fig. 12).

In addition to known DSB repair proteins (PARP1, BRCA1) we identified two novel proteins capable of influencing DNA repair outcomes: calcium voltage-gated channel subunit alpha1 C (CACNA1C) and aldehyde oxidase 1 (AOX1) (Fig. 3e). Silencing of *CACNA1C*, which was associated with HDR inhibition in STRING, increased HDR when used alone, and had an additive effect when combined with M3814 for both lipofection of oligos into iCas9 cells (Fig. 3f) and electroporation of Cas9-HiFi RNP for *FRMD7*, but hardly increased HDR for *NOVA1* (Fig. 3g). Transient *AOX1* knock-down further increased HDR when combined with M3814 (*FRMD7/NOVA1*: 1.2/1.3-fold), but not alone (Fig. 3e, g). In line with knock-down by siRNA, treatment with the calcium channel blocker drug nicardipine[55] or AOX1 inhibitor drug raloxifene[56] also showed a trend towards increased HDR when combined with M3814 (Fig. 3h).

Previously, we demonstrated that MMEJ inhibitors preferentially increase HDR when NHEJ is already inhibited[21]. Based on its STRING association with MMEJ enhancement (Supplementary Data 4) and a comparable repair outcome to the silencing of the MMEJ repair gene *POLQ* (Fig. 3e), we hypothesized that AOX1 is either associated with or may affect MMEJ. To test whether this effect extends to genetically inactivated *AOX1*, we generated a single cell-derived knock-out line and performed genome editing of *FRMD7* with or without NHEJ inhibition by M3814 (Supplementary Fig. 13). When combined with M3814, HDR and MMEJ were reduced by half, accompanied by 90% cell death, suggesting synthetic lethality when both DNA-PKcs and AOX1 are inhibited (Supplementary Fig. 13). Strikingly, bioluminescence strongly increased in AOX1-deficient cells for the HR (3-fold), NHEJ (9-fold), and MMEJ (6-fold) extrachromosomal repair reporter assays compared to wild type cells (Supplementary Fig. 14).

## Mechanistic insights of ESR2 and AOX1 in DSB repair

After having discovered the impact of *ESR2* and *AOX1* inhibition on DNA DSB repair outcomes, we set out to explore possible mechanisms. *ESR2* has been reported to suppress DNA damage-induced expression of the key HR genes breast cancer-related gene 1/2 (*BRCA1/2*)[57,58]. To test this, we quantified their expression in wild type and *ESR2* knock-out cell lines with and without DSB-inducing cisplatin treatment. *BRCA1* and *BRCA2* mRNA levels were upregulated in *ESR2* knock-out cells for both conditions (1.4- and 1.2-fold with cisplatin) (Fig. 4a). *ESR2* knock-out cells showed increased p-BRCA1 (S1524) and BRCA2 protein levels with cisplatin (2.2- and 1.3-fold), but not in the untreated condition (Fig. 4b). We then quantified four additional key DNA DSB repair proteins (ATM, Polθ, DNA-PKcs, LIG4) as well as a stress-responsive chaperone (HSP27). We found that even under regular cell culture conditions, ATM and DNA-PKcs protein levels were increased in *ESR2* knock-out cells (1.8- and 1.4-fold), while LIG4 levels were decreased by 30%. Polθ and HSP27 levels were comparable to wild type cells. AOX1-deficient cells showed largely similar changes in p-BRCA1, ATM, and DNA-PKcs (Fig. 4c). In contrast, however, we found that HSP27 protein levels were strongly reduced by 70% in these cells. All relative protein changes with or without cisplatin treatment are summarized in heatmaps for ESR2- (Fig. 4d) and AOX1-deficient (Fig. 4e) cells.

To further advance our mechanistic understanding of the impact of ESR2 and AOX1 on cellular DNA repair, we treated wild type, ESR2-, and AOX1-deficient cells with cisplatin or doxorubicin and stained them for γ-H2AX and 53BP1 (Fig. 4f, g and Supplementary Fig. 15–17). Under untreated conditions, γ-H2AX foci did not increase; however, 53BP1 foci were 2.6- and 5.6-fold more frequent in ESR2- and AOX1-deficient cells, respectively, compared to wild type cells. Notably, the levels of 53BP1 in untreated knock-out cells are nearly equivalent to those observed after 6 h of cisplatin treatment in wild type cells. Doxorubicin induced more 53BP1 foci than cisplatin across all genotypes. Across tested conditions, we observed more 53BP1 foci in both ESR2- and AOX1-deficient cells compared to wild type cells.

AOX1 is an enzyme that catalyzes the oxidation of aldehydes and has been described to play a role in reactive oxygen species (ROS) homeostasis[59]. ROS are highly reactive molecules that can induce DNA damage. Therefore, we investigated the effect of *AOX1* knock-out on hydrogen peroxide-induced ROS using an established assay that measures total cellular ROS by the oxidation of 2′,7′-dichlorodihydrofluorescein diacetate (DCFDA). We found that relative to wild type cells, ROS production was reduced in AOX1-deficient cells by a quarter (Supplementary Fig. 18a,b). However, after cisplatin treatment, which induces both ROS and DNA damage, we observed that the cell viability of AOX1-deficient cells was halved when compared to wild type cells (Supplementary Fig. 18c).

## Novel synthetic lethality drugs

Due to the synthetic lethality of MMEJ and HR, PARP inhibitors that block MMEJ are effective against many HR-deficient cancers[31]. Recently, we and others have also identified synthetic lethality of MMEJ and NHEJ[21,60] (Fig. 5a). Given that DNA repair genes are frequently mutated in cancer, and that mutations in the major DSB repair pathway regulators—*BRCA1/2* (HR), *PRKDC* (NHEJ), and *POLQ* (MMEJ)—account for 11% of all cancers[61] (Fig. 5b), we set out to investigate the potential of repurposed drugs to exhibit synthetic lethality. Using CRISPR editing, we generated hiPSCs deficient in NHEJ (DNA-PKcs K3753R) and attempted to knock-out *BRCA2*. Since we could not generate a proliferating knock-out of *BRCA2* in hiPSCs, we decided to use *BRCA2*-negative Capan-1 cells as model for HR deficiency[62]. Since synthetic lethality can be enhanced by chemotherapeutic agents that induce DSBs, we tested different concentrations of cisplatin for the above cell lines and determined the respective EC75 (concentration that induces 25% cell death) for subsequent experiments. hiPSCs were 3.3-fold more sensitive to cisplatin compared to Capan-1 cells (Fig. 5c).

Drugs with a potential to disrupt MMEJ could be synthetically lethal to both NHEJ- and HR-deficient cells. We thus tested synthetic lethality using artemether, which we identified as MMEJ outcome inhibitor in our drug screen (Fig. 2c, d), as well as raloxifene, erlotinib, and gefitinib, which have been described to inhibit AOX1[63]. Dose-responses of all these drugs showed synthetic lethality in NHEJ-deficient cells, even when no cisplatin was added (Fig. 5d). To assess synthetic lethality in leukemia eHAP1 cells, we used CRISPR editing to generate an isogenic NHEJ-deficient eHAP1 cell line carrying DNA-PKcs K3753R. In eHAP1 cells, both erlotinib and raloxifene also show synthetic lethality, while gefitinib and artemether do not (Supplementary Fig. 19). In addition to synthetic lethality in NHEJ-deficient cells, we also observed synthetic lethality in HR-deficient Capan-1 cells after treatment with raloxifene when compared to wild type hiPSCs (Fig. 5e). To exclude a non-isogenic artefact of comparing stem cells and Capan-1 cells, we used CRISPR editing to revert the *BRCA2* 6174delT mutation in Capan-1 cells, generating a single cell-derived line with restored functional *BRCA2*. We then tested the effect of the above drugs, as well as the PARP inhibitor olaparib, on the survival of HR-deficient and HR-proficient Capan-1 cells. All of the FDA-approved drugs tested here showed synthetic lethality in HR-deficient cells (Fig. 5f). Artemether

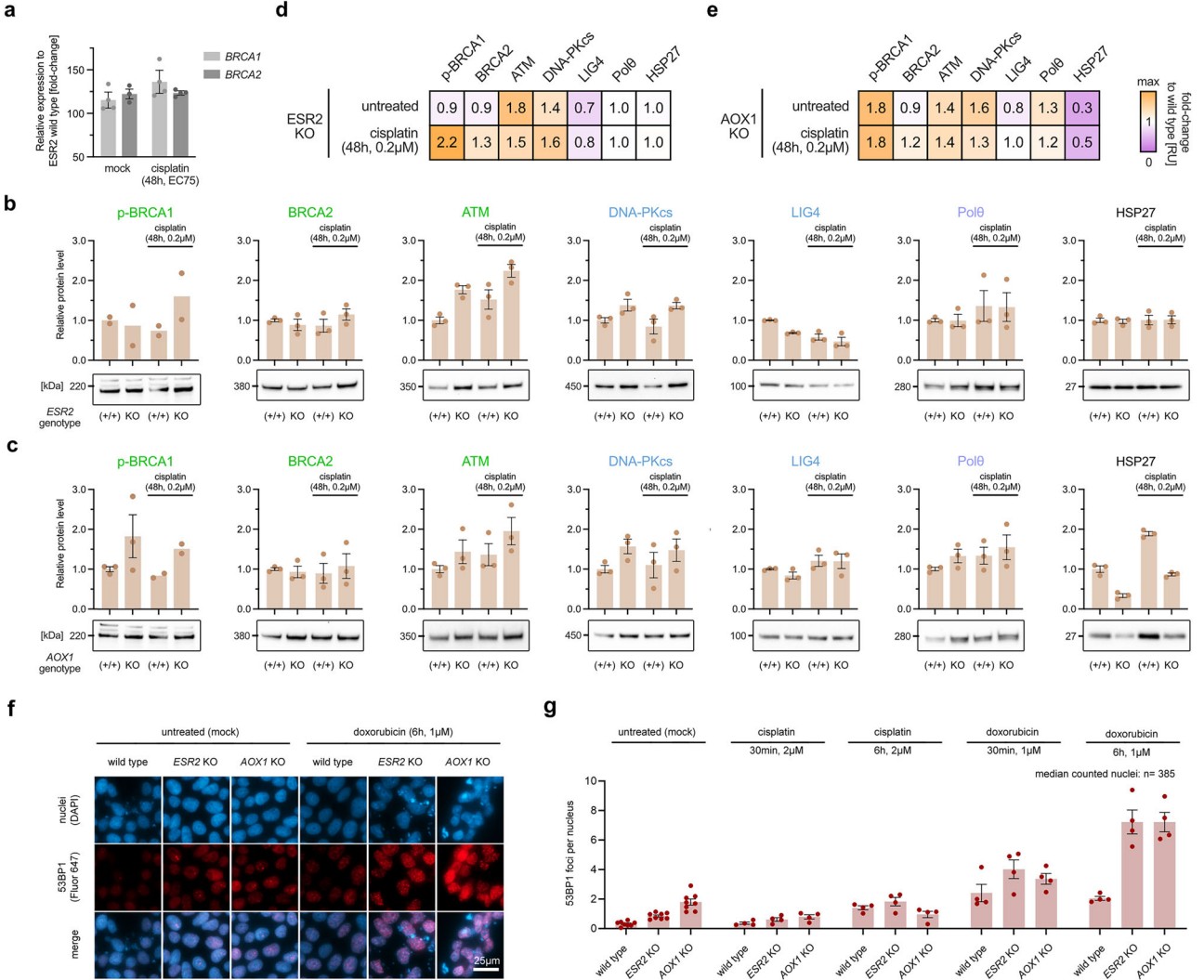

**Fig. 4 | Mechanistic insights of ESR2 and AOX1 in DSB repair. a** Relative expression levels of *BRCA1/2* in hiPSCs with biallelic *ESR2* knock-out when compared to wild type controls. Cells were either untreated (mock) or treated with cisplatin (37 nM, EC75) for 2 days. **b** Western blots and quantification of key DNA repair proteins associated with HR (p-BRCA1 [Ser1524], BRCA2, ATM), NHEJ (DNA-PKcs, LIG4), MMEJ (Polθ), and stress (HSP27) in wild type and *ESR2* knock-out 409B2 hiPSCs, with or without cisplatin treatment (48 h, 0.2 μM). **c** Western blots and quantification of key DNA repair proteins in wild type and *AOX1* knock-out 409B2 hiPSCs, with or without cisplatin treatment (48 h, 0.2 μM). **d** Heatmap summarizing (**b**). **e** Heatmap summarizing (**c**). **f** Representative immunostaining images of 53BP1 (Fluor 647) and nuclei counterstain of wild type, *ESR2* knock-out,

and *AOX1* knock-out cells after treatment with and without 1 μM doxorubicin for 6 h. Images were equally increased for brightness and contrast. Unmodified images without magnification are shown in Supplementary Fig. 15–17. These include treatment of cisplatin and doxorubicin for 30 min and 6 h, and also contain γ-H2AX staining (Fluor 488). **g** Quantification of 53BP1 foci per nucleus corresponding to **f** and Supplementary Fig. 15–17. Dots indicate counts from different from independent replicates, and the median number of counted nuclei across all conditions is stated. Independent biological replicates were performed (*n* = 4 for **a**, *n* = 3 for **b**–**e** except *n* = 2 for p-BRCA1 in **b**, *n* = 8 for untreated cells in **g**, *n* = 4 for treated cells in **g**). Error bars indicate the s.e.m. Source data are provided as a Source data file.

and the AOX1 inhibitors erlotinib and gefitinib reduced survival of HR-deficient cells (53–59% decrease) to a comparable extent as olaparib (67% decrease) at concentrations that did not reduce survival of HR-proficient cells.

Finally, we combined genome editing and synthetic lethality by reverting the DNA-PKcs K3753R mutation, a genetic model for *PRKDC* loss-associated immunodeficiency[64], back to the wild type using CRISPR Cas9-HiFi RNP editing and subsequent culturing in media containing our identified repurposed MMEJ inhibitors (Fig. 5g). Raloxifene increased the correction to the wild type state from 41% to 48% during editing. After two weeks, culturing in media containing artemether or raloxifene enriched corrected cells to 84% or 90%, respectively (mock 67%), presumably due to synthetic lethality in unedited NHEJ-deficient cells.

## Discussion

Every cell in the human body faces the threat of naturally occurring DNA DSBs[1]. DSB repair pathways are therefore essential for preventing cell death, and DSB repair pathway choice is of great importance in genome editing and synthetic lethality (Supplementary Fig. 1). Consequently, clinically safe drugs that can alter repair pathway choice would be of great benefit for gene therapy and oncology. Since the development of new pharmaceutical drugs is a time-consuming and costly process, taking 10–15 years and an average cost of over $1–2 billion for each new drug to be approved[65–67], we set out to repurpose clinically safe drugs by investigating the effect of most FDA-approved drugs on DSB repair pathway choice after introduction of a targeted DSB by CRISPR Cas9. Of note, our observations are limited to repair of targeted blunt-end DSBs, but drugs might have similar effects on DSBs

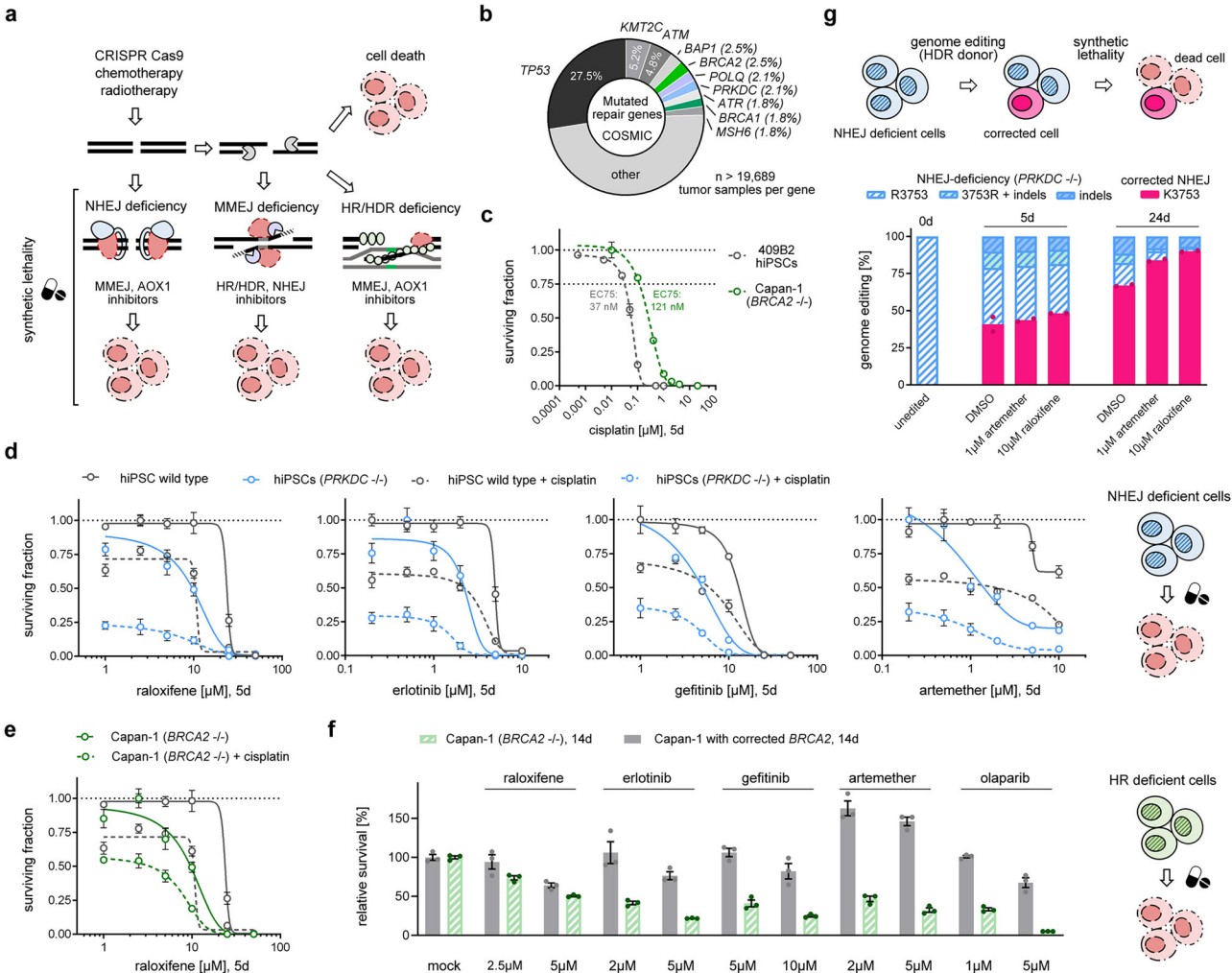

**Fig. 5 | Synthetic lethality by drugs repurposed as DSB repair pathway outcome inhibitors. a** Schematic of synthetic lethality of DSB repair pathways. DNA DSBs can be utilized to kill cancer cells that carry a genetic deficiency in DSB pathways when also provided with matching DSB pathway inhibitor drugs. **b** Percentage of top ten frequently mutated DNA repair genes in common cancers from the COSMIC database (data from Chae et al.[61]). Genes involved in HDR, MMEJ, and NHEJ are shaded green, purple, and blue, respectively. **c** Dose-response curve of surviving fraction of cells for treatment of 409B2 hiPSCs or Capan-1 cells with the DSB-inducing drug cisplatin for 5 d. The respective EC75 (concentration that induces 25% cell death) is stated. **d** Dose-response curves of surviving fractions in wild type (grey) and *PRKDC*-inactivated 409B2 hiPSCs (blue), with (dashed lines) or without (solid lines) cisplatin treatment for 5 days. Sigmoidal fits are shown as lines. **e** Dose-

response curves of surviving fractions in wild type 409B2 hiPSCs (grey) and *BRCA2*-inactivated Capan-1 cells (green), with (dashed lines) or without (solid lines) cisplatin treatment for 5 days. Sigmoidal fits are shown as lines. **f** Relative survival of *BRCA2*-corrected (grey) and *BRCA2*-inactivated (green) Capan-1 cells treated with drugs for 14 days, compared to the solvent control. **g** Genome editing efficiencies of correction of the DNA-PKcs K3753R mutation back to the wild type state using CRISPR Cas9-HiFi RNP editing in 409B2 hiPSCs and subsequent culturing in media containing no drug, artemether, or raloxifene. For corrected cells (pink), replicates are depicted by dots. Independent biological replicates were performed (*n* = 4 for **c**; *n* = 3 for **d**–**f**; *n* = 2 for **g**). Error bars show the s.e.m. Source data are provided as a Source data file.

introduced by collapsed DNA replication forks, or exposure to ionizing radiation, or other damaging events[1].

We used human iPSCs that have a diploid karyotype and do not already carry mutations affecting DNA repair, evaluated drug concentrations over several orders of magnitude to ensure the detection of potential biological effects (>7000 conditions), and provided a multidimensional readout that includes information on cell survival and Illumina sequencing-based repair pathway assignment of genome editing events (HDR, NHEJ, MMEJ) (Fig. 1b, c). The value of the multidimensional readout is illustrated by the screen's ability to detect the drug ouabain, a cardiac glycoside that is in clinical use for the treatment of heart failure, as a potent MMEJ enhancer, with the strongest relative change (6-fold MMEJ increase) in the entire dataset (Fig. 1d, g). To the best of our knowledge, no enhancers of MMEJ have been previously described. Ouabain also reduces NHEJ, HDR, and cell survival. Among the drugs in two previous screens, ouabain showed the most

potent phenotype in inhibiting DSB repair, but the assays used could only assess NHEJ and HR, leading to the conclusion that it would be a general inhibitor of DSB repair[68,69]. Our observed increase in MMEJ is consistent with the observation that ouabain appears to inhibit the retention of the pro-NHEJ factor 53BP1 at the site of DSBs[68].

Among the 22 repurposed DSB repair modulator drugs are PARP inhibitors, which are known inhibitors of MMEJ[12]. Olaparib inhibited MMEJ to a similar extent as genetically inactivated MMEJ by knock-out of *POLQ*[21]. Without normalization to MMEJ decreases observed in a *POLQ* knock-out, outcomes attributed to MMEJ are only reduced by a quarter, due to the limited precision in scoring of NHEJ- and MMEJ-derived indels, and low initial MMEJ outcomes when NHEJ is not blocked[3]. With the exception of PARP inhibitors and ouabain, the repurposed DSB repair outcome modulator drugs (Fig. 2c) have not previously been implicated in altering DNA DSB repair pathway outcomes, but for many, there is some evidence for their involvement in

DNA damage or repair[70–78]. The increase in MMEJ induced by dulox-etine may occur as a downstream consequence of nicotinic receptor inhibition[79], as suggested by a similar increase of MMEJ by a knock-out of the nicotinic receptor subunit *CHRNA5* (Figs. 2c, 3a). The reduction of HDR and MMEJ and increase in NHEJ outcomes by the nucleoside isomer cytarabine is likely due to termination of synthesized DNA strands and reliance on DNA synthesis (Supplementary Fig. 6). The extrachromosomal reporter assays confirm reduction of HR and MMEJ by cytarabine, but also show reduced NHEJ. In these assays, repair pathways do not compete with each other. The increase in NHEJ in the endogenous system, which is contrary to the decrease of NHEJ in the reporter assay, is likely due to compensation for inhibited HDR and MMEJ by the dominant end-joining pathway NHEJ. This shows that, although extrachromosomal reporter assays can measure the indivi-dual activity of repair pathways, they are not good at representing competition between intracellular pathways.

The identified cell survival enhancer, myristic acid, has been described to reduce the expression of the apoptosis proteins p21 and Bax in response to DNA breaks in primary mouse embryonic fibroblasts and osteoblasts[80]. Compared to the initial screen, repeated editing with a subset of drugs (Fig. 2b, c) resulted in reduced effects of cell survival enhancers (from max. 4.2-fold to 1.5-fold), as well as cell death by topoisomerase inhibitors at lower concentrations. High cell death is associated with lowered genome editing efficiencies (Fig. 1g) and is likely the reason for the reduced HDR-enhancing effects of topoisomerase inhibitors. This difference in susceptibility for cell death might be due to higher cell passage number and/or cell culture media batch effects[81] and calls for careful titration of drug concentrations in the target cells of choice.

Recently, a CRISPR screen produced a high-resolution atlas of the genetic dependencies of DSB repair outcomes[17]. However, this approach was limited to prior knowledge of around 500 genes involved in DSB repair or associated processes. The unbiased nature of screening drugs, which could in principle interact with any proteins expressed in the cell, allowed us to identify genes and processes involved in modulating DSB repair outcomes that are absent in DNA repair and damage response databases[82,83]. Surprisingly, cholinergic receptors, which play an important role in neural transmission within the nervous system, are targets for increasing cell survival and genome editing, and specifically HDR efficiency (Figs. 1f and 2e). Hinting to a more active DNA damage response, we find that orphenadrine, an inhibitor of cholinergic receptors, triples 53BP1 foci per nuclei (Sup-plementary Fig. 9). Cholinergic receptors have also been shown to be involved in viability and proliferation of stem cells[84,85]. It is conceivable that this change in proliferation alters levels of HR proteins[86].

Isogenic knock-out or siRNA-based silencing of *ESR2* can increase HDR outcomes in genome editing events. Strikingly, the combination of both inhibition of competing NHEJ by M3814 and active promotion of HDR by siRNA targeting *ESR2* can synergistically improve HDR outcomes (Fig. 3e). ESR2 inhibition likely increases HDR via increased levels of central HR proteins BRCA1/2 and ATM, and decreased levels of the NHEJ protein LIG4 (Fig. 4d). However, we also observe an increase in DNA-PKcs, which is central to NHEJ. Compatible with the increased levels of ATM, which phosphorylates 53BP1 to form discrete nuclear foci[87], we observe increased 53BP1 foci per nucleus—even in the absence of DNA damage-inducing agents (Fig. 4f, g). The syner-gistic effect of combining ESR2 inhibition with M3814, could be due to both increased levels of HDR proteins and inhibition of the kinase domain of DNA-PKcs, while the increased levels of the full DNA-PKcs protein may increase the amount of ATM needed for HDR[88]. In the absence of DNA-PKcs protein, levels of ATM, and thus HDR, are reduced[22]. Our findings add to the evidence that estrogen receptors influence the DNA damage response[89], making them promising phar-macological targets. In addition, silencing of the L-type calcium channel subunit *CACNA1C* can also increase HDR (Fig. 3e, f), and it is acknowledged that ion channels can have a multifaceted impact on DNA damage response[90].

In contrast to silencing of *ESR2* or *CACNA1C*, but similar to the MMEJ gene *POLQ*, silencing of AOX1 only increases HDR when NHEJ is blocked (Fig. 3e, g). The primary function of AOX1 is the oxidation of aldehydes to carboxylic acid[91], but both transient silencing and iso-genic knock-out (Fig. 3e and Supplementary Fig. 13) suggest an involvement in DNA damage response and repair. We find that pro-tein levels of ATM, p-BRCA1, DNA-PKcs, and Polθ are increased in AOX1-deficient cells compared to wild type cells (Fig. 4c), suggesting more active repair by HDR, NHEJ, and MMEJ. In line with this, extra-chromosomal bioluminescence reporter assays also show increased activity of HR, NHEJ, and MMEJ (Supplementary Fig. 14). Compatible with increased ATM levels and similar to *ESR2* knock-out cells, we also observe an increase in 53BP1 foci per nucleus in AOX1-deficient cells (Fig. 4f, g). Contrary to ESR2-deficient cells, AOX1-deficient cells have strongly reduced levels of the stress-responsive chaperone HSP27 (Fig. 4c), which could be a consequence of the observed reduced ROS levels (Supplementary Fig. 18). It is conceivable that even with lower ROS levels, deficiency to oxidate aldehydes could result in aldehyde accumulation and increased aldehyde-derived DNA adducts[92,93], which in turn activates DNA damage signaling and repair pathways.

Raloxifene, erlotinib, and gefitinib, which have been shown to inhibit AOX1[63], as well as artemether, which we identified as an MMEJ outcome inhibitor, exert synthetic lethality in *PRKDC*-deficient iPSCs and in *BRCA2*-deficient Capan-1 cells (Fig. 5d–f), making them inter-esting drug candidates for the treatment of NHEJ- as well as HR-deficient cancers. Strikingly, they show comparable performance to widely used olaparib and could be effective in PARP inhibitor-resistant HR-deficient cancers[94]. For NHEJ-deficient eHAP1 leukemia cells, only raloxifene and erlotinib resulted in synthetic lethality (Supplementary Fig. 19), which is compatible with that cell type-dependent variation can affect synthetic lethality[95]. More than 4% of all cancers have mutations in either *PRKDC* (NHEJ) or *BRCA1/2* (HR)[61], representing approximately one million new cancer patients world-wide every year[96]. The synthetic lethality of DNA-PKcs and AOX1 identified in this study also suggests a potential treatment for blad-der cancer with epigenetic loss of *AOX1*[97] using DNA-PKcs inhibitors. Raloxifene is known to increase survival of breast cancer patients, and this has been attributed to its inhibition of estrogen-stimulated growth of cancer cells[98]. Our findings suggest a second mechanism in which breast cancer cells, which are often deficient in the HR genes *BRCA1/2*, are killed due to synthetic lethality of raloxifene-induced inhibition of AOX1. Intriguingly, this synthetic lethality could also be used to enrich genetically corrected cells following ex vivo editing cells of patients suffering from NHEJ deficiency disorders[64,99], as we showed for a model of reverting *PRKDC* deficiency (Fig. 5g). This also implies that drugs identified as MMEJ outcome inhibitors as well as *AOX1* inhibitors should be contraindicated in patients with inherited NHEJ deficiency. Finally, it is tempting to speculate that HDR inhibi-tors identified in our screen could be synthetically lethal in MMEJ-deficient cancer cells, such as those deficient in *POLQ*, which exhibit ultra-high mutation rates[100].

In summary, we (1) provide an atlas of DSB repair response effects for most FDA-approved drugs, (2) identify clinically safe inhibitors and enhancers of repair pathway-dependent mutational outcomes attributed to NHEJ, MMEJ, and HDR, and (3) discover genes involved in DNA repair that may serve as future drug targets. In addition to applications in disease modeling, potential gene therapy, and oncology, the atlas may also be a valuable resource for clinicians, as the effect of some drugs on DSB repair might cause unintended side effects, thus helping to tailor drug treatments to an individual's genetic makeup.

## Methods

### Small molecules and oligonucleotides

For initial drug screening, the DiscoveryProbe FDA-Approved Drug Library (ApexBio, catalog no. L1021, as of November 2, 2021) was used, comprising 2344 FDA-approved small molecules and active pharmaceutical ingredients. By the end of 2013, 1453 small molecule drugs were approved[42], with 328 added between 2014 and 2023, totaling 1781 by the end of 2023[43]. For further drug testing, ART558 (MedChemExpress, catalog no. HY-141520), artemether (MedChemExpress, catalog no. HY-N0402), B02 (Sigma, catalog no. HY-101462), bromfenac sodium (MedChemExpress, catalog no. HY-B1888A), cisplatin (MedChemExpress, catalog no. HY-17394), doxorubicin (MedChemExpress, catalog no. HY-15142A), cyproterone acetate (MedChemExpress, catalog no. HY-13604), cytarabine HCl (MedChemExpress, catalog no. HY-13605A), dantrolene sodium salt (ApexBio, catalog no. B6329), dronedarone HCl (MedChemExpress, catalog no. HY-75839), duloxetine HCl (MedChemExpress, catalog no. HY-B0161A), erlotinib HCl (Sigma-Aldrich, catalog no. SML2156), gefitinib (Sigma-Aldrich, catalog no. SML1657), hodostin (neostigmine methyl sulfate; MedChemExpress, catalog no. HY-B1206), isosorbide 5-mononitrate (MedChemExpress, catalog no. HY-B0642), loperamide HCl (MedChemExpress, catalog no. HY-B0418A), M3814 (MedChemExpress, catalog no. HY-101570), myristic acid (MedChemExpress, catalog no. HY-N2041), olaparib (MedChemExpress, catalog no. HY-10162), ouabain octahydrate (MedChemExpress, catalog no. HY-B0542), raloxifene HCl (Sigma-Aldrich, catalog no. 1598201), rucaparib (MedChemExpress, catalog no. HY-10617A), were used. All gRNAs, DNA donors, primers, and siRNAs were ordered from Integrated DNA Technologies (Supplementary Data 5).

### Cell culture

We used 409B2 hiPSC (Riken BioResource Center, catalog no. HPS0076, GMO permit AZ 54-8452/26) containing iCRISPR Cas9[27]. Cells carrying a DNA-PKcs K3753R mutation were generated using CRISPR HDR based genome editing[21]. Cells were cultured at 37 °C in a humidified incubator with 5% $CO_2$ on Matrigel Matrix (Corning, catalog no. 35248) coated plates in mTeSR1 medium (StemCell Technologies, catalog no. 05851) with supplement (StemCell Technologies, catalog no. 05852) and daily media change. At ~80% confluence, stem cells were dissociated using EDTA (VWR, catalog no. 437012C) and reseeded at a split ratio ranging from 1:6 to 1:10 in medium supplemented with 10 µM Rho-associated protein kinase (ROCK) inhibitor Y-27632 (Calbiochem, catalog no. 688000) for one day after replating. Capan-1 cells (CLS Cell Lines Service, catalog no. 300143) were grown in KnockOut DMEM (Gibco, 10829018) supplemented with 2 mM GlutaMAX (Gibco, 35050061) and 10% FBS. The cell line THP-1 (Cytion, catalog no. 300356) was cultured in RPMI 1640 (ThermoFisher, catalog no. 11875-093) with 10% FBS. HEK293 cells (Cytion, catalog No. 300192) were grown in Dulbecco's modified Eagle's medium/F-12 (Gibco, catalog no. 31330-038) supplemented with 10% fetal bovine serum (FBS) (SIGMA, catalog no. F2442) and 1% NEAA (SIGMA, catalog no. M7145). Haploid eHAP1 cells (Horizon, catalog No. C669) and immortalized myelogenous leukemia cells (K562) (ECACC, catalog no. 89121407) were grown in Iscove's modified Dulbecco's media (ThermoFisher, catalog No. 12440053) with 10% FBS. Media was replaced every second day, and cells were split 1:6 to 1:10 once per week. The authenticity of the cell lines was confirmed both by the supplier via certificate of analysis and in-house through morphological examination. All cell lines were tested negative for mycoplasma contamination before and after the experiments.

### Electroporation of oligonucleotides

Stem cells were treated with TrypLE (Gibco, catalog no. 12605010) for 5 min at 37 °C and then triturated to acquire single cells. Following this, preheated media was added, and cell counts were determined using the Countess Automated Cell Counter 3 (Invitrogen). Subsequently, the cell suspensions were centrifuged at $300 \times g$ for 3 min at 25 °C. Electroporation was carried out using the B-16 program of the Nucleofector 2b device (Lonza) with cuvettes containing 100 µl Human Stem Cell nucleofection buffer (Lonza, catalog no. VVPH-5022), 1 million cells, 100 pmol electroporation enhancer, 320pmol gRNA (crRNA/tracR duplex for Cas9), and 200pmol of single-stranded DNA donor.

### Lipofection of oligonucleotides

409B2 iCRISPR Cas9 hiPSCs were incubated in medium containing 2 µg/ml doxycycline (Clontech, catalog no. 631311) 3d before lipofection to induce Cas9 expression. Lipofection was carried out using a final concentration of 7.5 nM of gRNA (crRNA/tracrRNA duplex), 7.5 nM of pre-designed siRNA, and 10 nM of the single-stranded DNA donor. In brief, the lipofection mix was prepared by separately diluting 0.75 µl of RNAiMAX (Invitrogen, catalog no. 13778075) and the respective oligonucleotides in 25 µl of OPTI-MEM (Gibco, catalog no. 1985-062), followed by 5 min incubation at 25 °C. Both dilutions were subsequently mixed to create a 50 µl solution of OPTI-MEM containing RNAiMAX, gRNAs, and the single-stranded DNA donor. The lipofection mix was incubated for 20–30 min at room temperature. Cells were dissociated using EDTA for 5 min and counted with the Countess Automated Cell Counter 3 (Invitrogen). The lipofection mix, 100 µl containing 25,000 dissociated cells in mTeSR1 supplemented with Y-27632, 2 µg/ml doxycycline, and either 2 µM M3814, 5 µM ART558, 10 µM B02, or no additional drug, was put in one well of a 96-well plate pre-coated with Matrigel Matrix. After 24 h, the medium was replaced with mTeSR1 with or without drugs, and after one additional day with mTeSR1 only.

### Resazurin viability assay

Following editing, cells were grown on clear flat-bottom black-wall 96-well cell culture plates (Greiner, catalog no. 655090, or Nunc, catalog no. 165305) and with media containing 10 µM ROCK inhibitor for 1d, followed by normal medium for 3d, before being transferred to fresh medium containing 10% resazurin solution (Cell Signaling, catalog no. 11884) and grown for 2 h before fluorescence reading using a CLARIOstar Plus microplate reader (BMG Labtech). Resazurin is converted by cellular dehydrogenases to the fluorescent resorufin, and the fluorescence (excitation: 530–570 nm, emission: 590–620 nm) reflects the amount of living cells[44]. Wells containing media and resazurin but without cells were used as blanks.

### Illumina library preparation

After at least five days post-transfection, cells were detached using TrypLE (Gibco, catalog no. 12605010), pelleted, and resuspended in 15 µl of QuickExtract DNA extraction solution (Lucigen, catalog no. QE09050). The samples were subjected to incubation at 65 °C for 10 min, followed by 68 °C for 5 min, and finally, 98 °C for 5 min to obtain single-stranded DNA. Subsequently, PCR was conducted in a T100 Thermal Cycler (Bio-Rad) using the KAPA2G Robust PCR Kit (Sigma, catalog no. KK5024) with buffer B and 3 µl of cell extract in a total reaction volume of 25 µl. The thermal cycling was: 95 °C 3 min; 34× (95 °C 15 s, 65 °C 15 s, 72 °C 15 s); 72 °C 60 s. Illumina adapters (P5 and P7) incorporating sample-specific indices were added in a subsequent PCR reaction[101], using Phusion HF MasterMix (NEB, catalog no. M0531) and 0.3 µl of the first PCR product. Cycling was: 98 °C 30 s; 25× (98 °C 10 s, 58 °C, 10 s, 72 °C 20 s); 72 °C 5 min. Amplifications were analyzed using 2% EX agarose gels (Invitrogen, catalog no. G4010–11), and indexed amplicons were purified using solid-phase reversible immobilization beads at a 1:1 ratio of beads to PCR solution[102]. Double-indexed libraries were sequenced on a MiSeq (Illumina) to produce paired-end sequences of $2 \times 150$ bp (+7 bp index). Following base calling with Bustard (Illumina), adapter sequences were removed using leeHom[103].

## Amplicon sequence analysis

Bam files were demultiplexed and converted into fastq files using SAMtools v.1.12[104]. Fastq files were then utilized as input for CRISPResso[105], which allowed for the analysis of the sequencing read percentages pertaining to various categories: wild type (unmodified), precise nucleotide substitution (HDR), indels (NHEJ and MMEJ), and a combination of both (mix). Analysis was limited to amplicons showing a minimum of 70% similarity to the wild type sequence and within a 20 bp window surrounding each gRNA. For an HDR event to be identified, sequence similarity threshold of 95% was set. Any unexpected substitutions were disregarded as potential sequencing errors. We employed a Python script to further identify sequencing reads with indels to be a likely result of NHEJ (<2 bp microhomology at deletion) or MMEJ (≥2 bp microhomology at deletion)[27].

## Extrachromosomal DNA DSB repair reporter bioluminescence assays

For siRNA-mediated knockdown, we used siRNAs against *PRKDC* (13.1, IDT), *POLQ* (13.8, IDT), *BRCA2* (13.1, IDT), or non-targeting siRNA (DS-NC1, IDT). Two days after seeding, the activity of double-strand break (DSB) repair pathways was assessed using a luciferase-based reporter system, with minor modifications from Rajendra et al. (2024). Reporter plasmids for Homologous Recombination (HR), Non-Homologous End Joining (NHEJ), and Microhomology-Mediated End Joining (MMEJ) were linearized and purified according to the reference protocol to create DSB repair substrates. For transfection, cells were co-transfected with a DSB repair reporter plasmid and a pGL4.53 Firefly luciferase plasmid (Promega) for normalization. In a 96-well plate, the amounts of DNA transfected per well were: 80 ng HR substrate and 2 ng pGL4.53 (HR), 20 ng NHEJ substrate and 27 ng pGL4.53 (NHEJ), and 80 ng MMEJ substrate and 20 ng pGL4.53 (MMEJ). Transfection was performed using ViaFect reagent (Promega, E4981) at a ratio of 4 μl reagent per 1 μg of total DNA. For each well, the DNA mixture and the ViaFect reagent were separately diluted in medium and incubated for 5 min. The two solutions were then combined, incubated for 20 min at room temperature, and 5 μl of the final transfection complex was added to the cells. Twenty-four hours post-transfection, luciferase activity was quantified using the ONE-Glo EX Luciferase Assay (Promega, E8150) on a CLARIOstarPlus microplate reader (BMG Labtech) according to the manufacturer's instructions. The data were normalized to the wild type condition with and without siRNA treatment.

## Quantitative PCR

Gene expression of *BRCA1/2* and the housekeeping gene *VCP* was analyzed using the TaqMan Fast Advanced Cells-to-Ct Kit (Invitrogen, catalog no. A35377). In summary, cells were lysed with genomic DNA removal by DNase I. RNA was reverse transcribed into cDNA (thermal profile: 37 °C 30 min, 95 °C 5 min), and quantitative PCR was done using CFX96 Real-Time-System C1000 Touch (Bio-Rad) with thermal profile: 50 °C 2 min, 95 °C 20 s, 40× (95 °C 1 s, 60 °C 20 s). For primers and probes, see Supplementary Data 5. Data was analyzed by the 2(-Delta Delta C(T)) method[106].

## Immunoblotting

Cultured cells were lysed using RIPA buffer (Thermo Scientific, catalog no. 89901) supplemented with Halt Protease Inhibitor Cocktail (Thermo Scientific, catalog no. 87786). Protein concentrations were determined with a BCA Protein Assay (Sigma-Aldrich, catalog no. BCA1-1KT). Depending on target protein weight, proteins (3–6 μg total protein) were electrophoresed according to the manufacturer's instructions on 4–12% NuPAGE Bis-Tris (Invitrogen, catalog no. NP0322) or 3–8% NuPAGE Tris-Acetate (Invitrogen, catalog no. EA03752), respectively. Gels were then transferred to a PVDF membrane (Invitrogen, catalog no. LC2002) using an XCell II Blot Module (Invitrogen, catalog no. EI9051). Total protein was fluorescently labeled using No-Stain Protein Labeling Reagent (Invitrogen, catalog no. A44449). After blocking with EveryBlot Blocking Buffer (Bio-Rad, catalog no. 12010020), the membranes were incubated with 1:1,000 dilutions of primary antibody overnight at 4 °C, respectively. Primary antibodies (all from Cell Signaling) were against targets ATM (catalog no. 2873), DNA-PKcs (catalog no. 4602), POLQ (catalog no. 64708), p-BRCA1 (catalog no. 9009), BRCA2 (catalog no. 10741), LIG4 (catalog no. 14649), HSP27 (catalog no. 2402). Then, the membranes were incubated with a 1:10,000 dilution of a horseradish peroxidase (HRP)-conjugated goat anti-rabbit IgG secondary antibody (Invitrogen, catalog no. 31460). Chemiluminescence signal was induced using Pierce ECL Western Blotting Substrate (Thermo Scientific, catalog no. 32209) or Super-Signal West Dura Extended Duration Substrate (Thermo Scientific, catalog no. 34075). Images were acquired using the iBright FL1500 Imaging System (Thermo Scientific) and analyzed using iBright Analysis Software (Thermo Scientific, version 5.4.0). Chemiluminescence band intensities were normalized to the total protein amount in each lane.

## Reactive oxygen species assay

Cellular reactive oxygen species (ROS) were measured using the DCFDA/H2DCFDA Cellular ROS Assay Kit (Abcam, catalog no. ab113851). Briefly, 20,000 cells were plated in clear, flat-bottom, black-wall 96-well plates (Greiner, catalog no. 655090). After 48 h, the cells were washed once with 1X Buffer (provided with kit), and incubated with 10 μM 2′,7′-dichlorofluorescin diacetate (DCFDA) for 45 min at 37 °C. Cells were then exposed to treatments of H2O2 or 75 μM tert-butyl hydroperoxide (TBHP) in culture media. ROS production was immediately determined by measuring the formation of the fluorescent compound 2′,7′-dichlorofluorescein using a CLARIOstar Plus microplate reader (BMG Labtech) with an excitation wavelength of 485 nm and an emission wavelength of 535 nm. Measurements were taken every 60 min for up to four hours. Cell viability was assessed in parallel using a Resazurin assay.

## Microscopy and image analysis

409B2 hiPSCs genotype variants were treated with DMSO solvent, 0.1 μM cytarabine, 2 μM M3814, 10 μM orphenadrine, 2 μM cisplatin, or 1 μM doxorubicin. Then, cells were stained for γ-H2AX, 53 BP1 using the PhenoVu DNA Damage Response Staining Kit (Revity, catalog no. PDDR11). In brief, cells were washed, fixed with 4% formaldehyde for 10 min at room temperature, and permeablized with 0.1% Triton X-100 for 10 min. Cells were then incubated for 1 h with primary antibodies (anti-Phospho H2AX (S139)−mouse IgG1, anti-53BP1−rat IgG2a), followed by incubation with secondary antibodies (Fluor 488−Goat anti-Mouse highly cross-adsorbed, Fluor 647−Goat anti-Rat highly cross adsorbed) and Hoechst 33342 to counterstain nuclei. After each of the above steps, cells were washed three times with DPBS. A fluorescent microscope Axio Observer Z (Zeiss) was used to obtain one image (objective LD Plan-Neofluar 20x/0.4 Korr Ph 2 M27), from each of four replicates for the respective treatments, consisting of the following: DAPI (BP 335−383 nm, BS 395 nm, BP 420−470 nm), Alexa Fluor 488 (BP 450−490 nm, BS 495 nm, BP 500−550 nm), and Cy5 (BP 625−655 nm, BS 660 nm, BP 665−715 nm). Images were blinded, and nuclei were counted using the Adobe Photoshop CS5 counting tool. 53BP1 foci were quantified using ImageJ (2.16.0/1.54p) using the difference of Gaussians.

## Hit category identification

To avoid false positive hit assignments caused by insufficient sequencing depth, drug conditions resulting in ≤1000 sequencing reads were filtered out. Boundaries for the metrics (HDR, NHEJ, MMEJ, and cell survival) deviating from the norm were established by calculating the 1st (Q1) and 3rd (Q3) quartiles relative to their non-drug controls (Supplementary Fig. 2c). We defined categories based on the DNA-damage response model (Fig. 1a). Each category indicates

whether the associated metric is allowed to increase (>Q3, indicated by "↗"), decrease (<Q1, indicated by "↘"), remain equal or increase (≥Q1, indicated by "→/↗"), or remain equal or decrease (≤Q3, indicated by "→/↘"). The identified categories are as follows: HDR enhancer (HDR ↗, NHEJ →/↘, MMEJ →/↘), HDR inhibitor (HDR ↘, NHEJ →/↗, MMEJ →/↗), NHEJ enhancer (HDR →/↘, NHEJ ↗, MMEJ →/↘), NHEJ inhibitor (HDR →/↗, NHEJ ↘, MMEJ →/↗), MMEJ enhancer (HDR →/↘, NHEJ →/↘, MMEJ ↗), MMEJ inhibitor (HDR →/↗, NHEJ →/↗, MMEJ ↘), cell survival enhancer (cell survival ↗). These criteria were subsequently applied in the primary drug screen. For further drug testing, a minimum of four best hits for each category were selected, and at least four of these had to be non-toxic (relative cell survival > Q1: 0.79). For dose-response curves we selected the lowest concentration of each drug that produced the desirable effect on DSB repair pathways in the initial screen and tested two higher and two lower additional concentrations (0.2-fold, 0.5-fold, 2-fold, and 5-fold).

### ChEMBL and STRING analysis of protein modulators in DSB repair pathways

The top 30 compounds from the primary drug screen were selected for enhancers and inhibitors of HDR, NHEJ, and MMEJ. For each compound, we used ChEMBL[52] to generate a list of the known primary protein target ("drug mechanisms"), other targets with published related bioactivity data ("target summary"; type = "single protein" organism = *Homo sapiens*), and "target predictions" (confidence levels = "active", organism = *Homo sapiens*). To control for the bias of the FDA-selected drugs, a random selection of 30 drugs was used to retrieve above-mentioned targets, which were subsequently subtracted from the data set of the selected compounds. The remaining targets were then clustered using STRING analysis (https://string-db.org/; v11.5[54]) to find connected proteins. For each cluster, we selected the protein with the highest number of nodes. For the interaction network of related bioactivity data, we selected a central protein of the cluster. We selected pre-designed siRNAs (Supplementary Data 5) for silencing of the related genes.

### Statistics and reproducibility

Bar graphs were generated, and standard errors of the mean (s.e.m.), Pearson correlations, and related *p* values were computed using GraphPad Prism 10 software. The figure legends specify the number of replicates. Sample size was not predetermined using a statistical method. The experiments were not randomized, and samples were prepared without blinding but in a parallel fashion. Data analysis was conducted based on numerical sample names, ensuring that the identity of the samples remained undisclosed during the analysis process.

### Reporting summary

Further information on research design is available in the Nature Portfolio Reporting Summary linked to this article.

## Data availability

The sequencing data generated in this study are deposited in the NCBI's Sequence Read Archive (SRA) with the accession code PRJNA1354267. Data are also deposited in the Dryad database under accession code dryad.1ns1rn970 [doi.org/10.5061/dryad.1ns1rn970]. Source data are provided with this paper.

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

## Acknowledgements

We thank S. Pääbo for helpful discussions and A. Weihmann and B. Schellbach for DNA sequencing. We thank Artios Pharma Limited for kindly providing DNA DSB repair reporter assay plasmids from Rajendra et al. 2024. Funding was provided by Merck Healthcare (Merck Biopharma Open Innovation Grant, S.R.), the Federal Ministry of Education and Research, Germany (GO-Bio initial Grant, S.R.), and the NOMIS foundation (S. Pääbo). Open access funding was provided by the Max Planck Society.

## Author contributions

S.R. conceived the idea. S.R., D.M., and P.K. planned and performed the experiments, analyzed the data, and wrote the paper.

## Funding

## Competing interests

The authors declare the following competing interests: Related patent application on repurposed drugs for DNA repair and increased prime editing efficiency (patent applicant: Max Planck Society; inventors: S.R., D.M., and P.K.; application number: PCT/EP2025/059927; status: pending).
