## [Transparent Peer Review file · Nature Communications]

Repurposing clinically safe drugs for DNA repair pathway choice in CRISPR genome editing and synthetic lethality

Corresponding Author: Dr Stephan Riesenberger

Version 1:

Reviewer comments:

Reviewer #1

(Remarks to the Author)

The manuscript investigates FDA-approved drugs for their ability to modulate DNA double-strand break (DSB) repair pathways, focusing on their effects on CRISPR genome editing outcomes and synthetic lethality. Using high-throughput screening in human induced pluripotent stem cells (hiPSCs), the authors evaluate over 2,000 drugs across multiple concentrations to identify modulators of non-homologous end joining (NHEJ), microhomology-mediated end joining (MMEJ), and homology-directed repair (HDR). Key findings include compounds that enhance or inhibit specific repair pathways and novel protein targets, such as ESR2, AOX1, and CACNA1C, implicated in repair modulation.

While ESR2 silencing is presented as an enhancer of HDR, its effect is modest, yielding only a 1.4- to 1.7-fold increase. In contrast, the primary driver of repair pathway modulation remains DNA-PK inhibition, a well-characterized mechanism that significantly overshadows the contribution of ESR2. Additionally, the high drug concentrations required to observe effects (e.g., 10 μ M) raise concerns about off-target effects and toxicity. For example, ouabain, identified as a potential MMEJ enhancer, causes over 90% cell death, limiting its therapeutic applicability. Novel targets like AOX1 and CACNA1C show limited independent activity, with significant effects only observed in combination with DNA-PK inhibitors, suggesting their contributions may be secondary or highly context-specific.

The reliance on high, toxic concentrations and modest, incremental effects undermines the study's translational relevance and originality. While the dataset provides breadth, it largely confirms existing knowledge rather than offering novel insights. The potential applications of these findings in gene editing or cancer therapy are constrained by these limitations, which significantly reduce the impact of this study.

Reviewer #2

(Remarks to the Author)

This manuscript presents a screening study focused on repurposing FDA-approved drugs to modulate DNA double-strand break (DSB) repair pathways. The authors screened over 7,000 drug conditions in human induced pluripotent stem cells and identified several compounds that can enhance or inhibit repair outcomes attributed to non-homologous end joining (NHEJ), microhomology-mediated end joining (MMEJ), and homology-directed repair (HDR) based on sequencing outcomes at a CRISPR targeted site and cell survival.

Key strengths of the study include: (1) The unbiased screening approach to evaluate FDA-approved drugs. (2) The identification of specific proteins with robust effects on DSB repair, including estrogen receptor 2 (ESR2) and aldehyde oxidase 1 (AOX1). (3) The study provides a valuable resource for the scientific community, including an atlas of inferred DSB repair effects based on DNA sequencing.

However, there are some limitations to consider: (1) The need for validation in other cell types. (2) Additional orthogonal assays to validate the key conclusions beyond the sequencing and survival outcomes. (3) A lack of mechanistic studies to elucidate how some of the identified compounds and proteins influence DSB repair. Inferences from gene editing or genome sequencing are insufficient. The key conclusions need to be supported by some mechanistic studies of HDR such as repair foci dynamics and DNA repair factor recruitment to sites of DSBs.

Overall, this manuscript presents a potential advance in our understanding of DSB repair modulation. However, without further validation and mechanistic confirmation it is premature to recommend publication in Nature Communications.

Version 2:

Reviewer comments:

Reviewer #1

(Remarks to the Author)

I want to thank the authors for the substantial effort they put into revising the manuscript and adding new experiments. The scale of the drug screen is impressive, and I can see that the authors have tried to respond carefully to the comments from the first round. That said, after going through the revised figures and the rebuttal in detail, I still have significant concerns.

The central point remains that the effect of ESR2 inhibition is not broadly convincing. The strongest result is the increase in HDR at the KNL1 locus with Cas12a, but at other targets, in other cell types, and with Cas9 editing, the impact of ESR2 knockout or knockdown is modest at best. In most cases, the reported gains are under two-fold, which does not support the idea that this approach will have a general role in improving gene editing outcomes.

The additional mechanistic data do not fully resolve this. The Western blots show differences that are quite small, generally less than two-fold, and difficult to interpret as strong functional effects. The immunostaining is even more problematic. Normally, pH2AX and 53BP1 foci correlate tightly, but in Figures 4f and 4g the images suggest an overall increase in 53BP1 staining without clear punctate foci. This makes the conclusion that ESR2 or AOX1 loss alters repair foci dynamics hard to accept.

The drug synergy claims are also not sufficiently supported. In Figure 5, synergy with cisplatin is shown at only a single concentration. Without testing across a range of doses and analyzing the interaction with accepted models of drug synergy, the conclusion remains tentative at best.

Taken together, while the dataset is large and some of the findings are interesting, the main results remain either context-dependent or incremental. The reliance on high or toxic drug concentrations further limits translational impact. The additional mechanistic experiments help to some degree but do not, in my view, provide strong new functional insights.

Reviewer #2

(Remarks to the Author)

The authors have responded appropriately to the prior review and the manuscript is now improved. I have no further concerns and am supportive of publication.

Version 3:

Reviewer comments:

Reviewer #1

(Remarks to the Author)

The authors have clarified several technical points. However, the main concerns remain. ESR2 effects are modest and context-dependent, mechanistic insights are limited, and drug synergy is not supported by rigorous analysis. Thus, the revisions do not fully resolve the issues raised in my previous review.

Reviewer #3

(Remarks to the Author)

I was requested to 'evaluate the concerns raised by reviewer #1 and indicate if the response from the authors is sufficient or not'.

I find the authors responded the points brought up by reviewer #1 in a way that I find sufficient overall and I support the publication.

Macak and colleagues performed a large screen of FDA approved drugs to look for molecules that affect DNA repair outcomes following gene editing in an iPSC line. This large screen produced a data set of useful concentrations for further screens and identified a number of small molecules and proteins affecting DNA repair- notably ESR2 and AOX1. A siRNA knockdown of ESR2 along with M3814 was shown to drive a notable increase in knock-ins. And initial investigations of the effects of AOX1 show it to be involved, or affect, DNA damage repair. Finally, the authors show drugs found by their screen to affect DNA repair can be used to induce synthetic lethality in their model system.

Reviewer #1 had five concerns/paragraphs responded to by the authors.

Point 1: 'ESR2 is not broadly convincing... with the impact modest at best'.

Reviewer #1 notes that the effect of ESR2 modulation on HDR is modest; while true I find the response that Figure 3 highlights the combined effects of DNA-PKcs & ESR2 inhibition, and together these effects are more dramatic, interesting, and likely of use for other to test in ones own system of interest.

Point 2: 'The additional mechanistic data do not fully resolve this ... and generally small... less than two-fold'

I agree with the authors that their data are "compatible" with increased HDR and that the data (Figure 4) generally support their claims. Reviewer #1's comment that the mechanisms are 'not fully resolved' is true, but I side with the authors that within the context of this paper, a more detailed mechanistic elucidation is outside the scope of the present work.

Point 3: Authors updated images and sup figures address this.

Point 4: 'The drug synergy claims are also not sufficiently supported. In Figure 5, synergy with cisplatin is shown at only a single concentration...'

While data on various concentrations of cisplatin would further strengthen the point, the author's response that the effect is seen in the knock out lines without cisplatin, and cisplatin only strengthens the effects is convincing.

Point 5: ' Taken together, while the dataset is large and some of the findings are interesting, the main results remain either context-dependent or incremental. The reliance on high or toxic drug concentrations limits translational impact.'

Between the sup data 2 and the 'new functional insights as AOX1's role in DNA damage repair and ESR2's influence on genome editing have not been reported previously' to be novel findings. Overall I side with the authors and find that their experimental design and findings from this large screen is an interesting and useful data set to publish.

Dear Reviewers,

We would like to thank you for the valuable input, and are very grateful for the time you have invested in improving our manuscript. Please find below a point-by-point response to your comments.

Sincerely,

Stephan Riesenber

Point-by-point-response

Reviewer #1:

The manuscript investigates FDA-approved drugs for their ability to modulate DNA double-strand break (DSB) repair pathways, focusing on their effects on CRISPR genome editing outcomes and synthetic lethality. Using high-throughput screening in human induced pluripotent stem cells (hiPSCs), the authors evaluate over 2,000 drugs across multiple concentrations to identify modulators of non-homologous end joining (NHEJ), microhomology-mediated end joining (MMEJ), and homology-directed repair (HDR). Key findings include compounds that enhance or inhibit specific repair pathways and novel protein targets, such as ESR2, AOX1, and CACNA1C, implicated in repair modulation.

While ESR2 silencing is presented as an enhancer of HDR, its effect is modest, yielding only a 1.4- to 1.7-fold increase. In contrast, the primary driver of repair pathway modulation remains DNA-PK inhibition, a well-characterized mechanism that significantly overshadows the contribution of ESR2.

We agree that DNA-PKcs inhibition has a strong and well described influence on the outcome of CRISPR editing. However, we show that ESR2 inhibition can also have strong effects. The 1.4- to 1.7-fold increase are from Fig. 3a and described a 1.4-fold increase in HDR and a 1.7-fold increase in survival for ESR2 knock-out cells compared to wild type iPSCs. Importantly, for transient inhibition of ESR2, we show a mean increase in HDR across several targets and cell types of 2.5-fold, while the DNA-PKcs inhibitor M3814 increases HDR by 2.1-fold (Fig. 3c). A combination with M3814 results in a synergistic increase (mean increase of 4.6-fold). This increase in HR repair pathway capabilities of ESR2-deficient cells was reproduced when we used an extrachromosomal DNA DSB repair reporter (Supplementary Fig. 11).

Additionally, the high drug concentrations required to observe effects (e.g., 10 μ M) raise concerns about off-target effects and toxicity. For example, ouabain, identified as a potential MMEJ enhancer, causes over 90% cell death, limiting its therapeutic applicability.

We agree with the reviewer that some of the drugs we have tested result in high cell death and would not be suitable for in vivo application, but might be relevant for cell line engineering We acknowledge this in the main text: 'Because short-term drug toxicity is tolerable for cell line

engineering in which surviving cells are propagated, we also tested combinations of drugs, independent of their toxicity, to explore potential additive effects...(Supplementary Fig. 7)'.

Nevertheless, almost all drugs selected for testing at other targets and cell lines (Fig. 2d, Fig. 5f, Supplementary Fig. 5) are active at concentrations of 2.5 μ M or well below (the only exception being 20 μ M duloxetine, and 10 μ M orphenadrine/tolterodine). One example of a likely non-toxic drug for in vivo use is erlotinib, which results in synthetic lethality in HR- and NHEJ-deficient cells (Fig. 5d, f, Supplementary Fig. 19), because it is effective at similar concentrations as the widely used breast cancer treatment drug olaparib (Fig. 5f).

Recognizing the importance of drug toxicity, we utilized our screening data to provide a single toxicity-reduced screening concentration for each drug for future FDA drug repurposing screens (Supplementary Data 2).

Novel targets like AOX1 and CACNA1C show limited independent activity, with significant effects only observed in combination with DNA-PK inhibitors, suggesting their contributions may be secondary or highly context-specific.

We agree with the reviewer that the effect of CACNA1C is highly context-specific, as significant effects are only observed in combination with DNA-PK inhibitors. We still believe that it adds valuable evidence to prior findings that ion channels can have multifaceted impact on DNA damage response.

However, our efforts to better understand the observed effects of AOX1, as well as ESR2, on CRISPR editing outcomes have led us to observe novel functions of these proteins in DNA DSB repair. We now used Western blotting to analyze the protein levels of six key DNA repair proteins involved in HR (ATM, p-BRCA1, BRCA2), NHEJ (DNA-PKcs, LIG4), and MMEJ (Pol θ), as well as a stress-responsive chaperone (HSP27) in wild-type versus *ESR2* knockout cell lines. Notably, we found that, even under regular culture conditions, the levels of ATM and DNA-PKcs proteins were higher in the *ESR2* knock-out cells (1.8- and 1.4-fold, respectively). Under stress induced by cisplatin, p-BRCA1 and BRCA2 levels were also higher in these cells (2.2- and 1.3-fold, respectively). Interestingly, we found largely similar effects in these proteins when comparing wild type and AOX-deficient cell lines. Additionally, AOX1-deficient cells showed strongly reduced HSP27 levels by 70%.

We further analyzed markers of DNA DSBs by immunostaining, and found that both *ESR2*- and AOX1-deficient cells had a higher number of 53BP1 foci per nucleus than wild type cells, consistent with the observed increased levels of ATM. This increase in foci is even stronger when cells are treated with cisplatin and doxorubicin, respectively.

We have presented these results as a new chapter in the main text and as a new Fig. 4 (with related Supplementary Fig. 9, 15, 16, 17). We have attached it also here for your convenience.

Fig. 4 | Mechanistic insights of ESR2 and AOX1 in DSB repair. (a) Relative expression levels of *BRCA1/2* in hiPSCs with biallelic *ESR2* knock-out when compared to wild type controls. Cells were either untreated (mock) or treated with cisplatin (37nM, EC75) for 2 days. (b) Western blots and quantification of key DNA repair proteins associated with HR (p-BRCA1 [Ser1524], BRCA2, ATM), NHEJ (DNA-PKcs, LIG4), MMEJ (Polθ), and stress (HSP27) in wild type and *ESR2* knock-out 409B2 hiPSCs, with or without cisplatin treatment (48h, 0.2μM). (c) Western blots and quantification of key DNA repair proteins in wild type and *AOX1* knock-out 409B2 hiPSCs, with or without cisplatin treatment (48h, 0.2μM). (d) Heatmap summarizing b. (e) Heatmap summarizing c. (f) Representative immunostaining images of 53BP1 (Fluor 647) and nuclei counterstain of wild type, *ESR2* knock-out, and *AOX1* knock-out cells after treatment with and without 1μM doxorubicin for 6h. Images were equally increased for brightness and contrast. Unmodified images without magnification are shown in Supplementary Fig. 15-17. These include treatment of cisplatin and doxorubicin for 30min and 6h, and also contain γH2AX staining (Fluor 488). (g) Quantification of 53BP1 foci per nucleus corresponding to f and Supplementary Fig. 15-17. Dots indicate counts from different from independent replicates and the median number of counted nuclei across all conditions is stated. Independent biological replicates were performed (n = 4 for a, n = 3 for b-e except n = 2 for p-BRCA1 in b, n = 8 for untreated cells in g, n = 4 for treated cells in g). Error bars indicate the s.e.m. Source data are provided as a Source Data file.

The reliance on high, toxic concentrations and modest, incremental effects undermines the study's translational relevance and originality. While the dataset provides breadth, it largely confirms existing knowledge rather than offering novel insights. The potential applications of these findings in gene editing or cancer therapy are constrained by these limitations, which significantly reduce the impact of this study.

We believe in the translational relevance and originality of our study, which is strengthened by substantial new data. Although we identified a few drugs with high toxic concentrations, we also describe drugs that could potentially be used for in vivo treatment, such as erlotinib for synthetic lethality. Additionally, transient *ESR2* inhibition may be a promising approach for cell line engineering and clinical ex vivo CRISPR applications. Further expanding the breadth of the dataset, we add novel mechanistic insights on the impact of *ESR2* and *AOX1* on DNA DSB repair.

Reviewer #2:

This manuscript presents a screening study focused on repurposing FDA-approved drugs to modulate DNA double-strand break (DSB) repair pathways. The authors screened over 7,000 drug conditions in human induced pluripotent stem cells and identified several compounds that can enhance or inhibit repair outcomes attributed to non-homologous end joining (NHEJ), microhomology-mediated end joining (MMEJ), and homology-directed repair (HDR) based on sequencing outcomes at a CRISPR targeted site and cell survival. Key strengths of the study include: (1) The unbiased screening approach to evaluate FDA-approved drugs. (2) The identification of specific proteins with robust effects on DSB repair, including estrogen receptor 2 (ESR2) and aldehyde oxidase 1 (AOX1). (3) The study provides a valuable resource for the scientific community, including an atlas of inferred DSB repair effects based on DNA sequencing.

Thank you for the kind words.

However, there are some limitations to consider: (1) The need for validation in other cell types. (2) Additional orthogonal assays to validate the key conclusions beyond the sequencing and survival outcomes. (3) A lack of mechanistic studies to elucidate how some of the identified compounds and proteins influence DSB repair. Inferences from gene editing or genome sequencing are insufficient. The key conclusions need to be supported by some mechanistic studies of HDR such as repair foci dynamics and DNA repair factor recruitment to sites of DSBs.

To strengthen our findings, we have addressed the mentioned points as follows:

- (1) We expanded the evaluation of potential DNA repair outcome modulators to additional cell lines. We tested selected compounds for CRISPR editing in three new cell lines (HEK293, K562, and THP1) and found comparable effects to those in hiPSCs. These findings are presented in Supplementary Fig. 5 and Supplementary Fig. 6a. Furthermore, we confirmed the effect on HDR increase upon silencing of *ESR2* in THP1 cells. We added the new data to Fig. 3c.

We also tested the synthetic lethality observed in NHEJ-deficient hiPSCs in a leukemia eHAP1 cell line by generating an isogenic line carrying DNA-PKcs K3753R. We found that both raloxifene and erlotinib exhibit synthetic lethality in the NHEJ-deficient eHAP1 cells, whereas gefitinib and artemether do not. This is compatible with that cell type-dependent variation can affect synthetic lethality. The results are presented in a new Supplementary Fig. 19.

- (2) To expand some of our findings with orthogonal assays, we used recently published reporter assays for HR, NHEJ, and MMEJ (Rajendra et al., 2024; PMID: 38109306). Briefly, these assays are based on repairing individually designed extrachromosomal DNA constructs for each pathway without competing with the other pathways. Upon successful repair, luciferase-induced bioluminescence can be measured.

Using these assays, we showed that the treatment of cells with cytarabine reduces their capability of DNA DSB repair for HR and MMEJ, confirming the effects observed after editing and sequencing in cells. However, we also observed that in the reporter, NHEJ

activity is reduced. We argue that the NHEJ increase in the endogenous system is likely due to compensation of inhibited HDR and MMEJ by the dominant end-joining pathway NHEJ. We discuss that although extrachromosomal reporter assays can measure the individual activity of repair pathways, they are limited at representing competition between intracellular pathways. The results are presented in a new Supplementary Fig. 6.

Applying the same reporter assays in *ESR2*- and *AOX1*-deficient cells showed that HR, NHEJ, and MMEJ-dependent bioluminescence is increased for both genotypes when compared to wild type cells. The results are presented in two new Supplementary Fig. 11 and 14.

- (3) To gain insights into the mechanisms of *ESR2* and *AOX1* in modulating DNA repair, we analyzed their influence on other DNA repair proteins and DNA repair foci dynamics. For *AOX1*, we also tested its role in reactive oxygen species homeostasis.

Specifically, we used Western blotting to analyze the protein levels of six key DNA repair proteins involved in HR (ATM, p-BRCA1, BRCA2), NHEJ (DNA-PKcs, LIG4), and MMEJ (Pol θ), as well as a stress-responsive chaperone (HSP27) in wild type versus *ESR2* knock-out cell lines. Interestingly, we found that, even under regular culture conditions, the levels of ATM and DNA-PKcs proteins increased in the *ESR2* knockout cells (1.8- and 1.4-fold, respectively). Under stress induced by cisplatin, p-BRCA1 and BRCA2 levels were also higher in these cells (2.2- and 1.3-fold, respectively). Interestingly, we found largely similar effects in these proteins when comparing wild type and *AOX1*-deficient cell lines. Additionally, *AOX1*-deficient cells showed strongly reduced HSP27 levels (0.7-fold).

To further investigate the effects of *ESR2* and *AOX1* on DNA repair we analyzed the abundance of nuclear DNA repair foci in these cell lines by staining them for γ -H2AX and 53BP1. We found that both *ESR2*- and *AOX1*-deficient cells exhibited more 53BP1 foci than wild type cells across all tested conditions – even in the absence of damage-inducing cisplatin or doxorubicin treatment.

We have presented these results as a new chapter in the main text and as a new Fig. 4 (with related Supplementary Fig. 9, 15, 16, 17). We have attached it also here for your convenience, see response to reviewer 1.

We add to the discussion how our new insights on *ESR2* influences DNA damage repair by modifying the levels of key HR proteins, such as ATM, which may subsequently modulate BRCA1/2, 53BP1, and DNA-PKcs. This adds to the idea that estrogen receptors are promising pharmacological targets for DNA damage repair and CRISPR editing.

Additionally, we investigated the effect of *AOX1* on reactive oxygen species (ROS), as these can cause cellular stress and DNA damage. Using an established assay, we found that relative to wild type cells, ROS production upon treatment with hydrogen peroxide was reduced in *AOX1*-deficient cells by ~30%. However, when treated with cisplatin, these cells were half as viable as wild type cells. These results are presented in a new Supplementary Fig. 18.

Overall, this manuscript presents a potential advance in our understanding of DSB repair modulation. However, without further validation and mechanistic confirmation it is premature to recommend publication in Nature Communications.

We also believe that this study advances our understanding of DSB repair modulation and that it has translational relevance. We are especially grateful for your suggestion to explore mechanistic insights, as it has added an exciting new dimension to our manuscript.

Reviewer #1 (Remarks to the Author):

I want to thank the authors for the substantial effort they put into revising the manuscript and adding new experiments. The scale of the drug screen is impressive, and I can see that the authors have tried to respond carefully to the comments from the first round. That said, after going through the revised figures and the rebuttal in detail, I still have significant concerns.

The central point remains that the effect of ESR2 inhibition is not broadly convincing. The strongest result is the increase in HDR at the KNL1 locus with Cas12a, but at other targets, in other cell types, and with Cas9 editing, the impact of ESR2 knockout or knockdown is modest at best. In most cases, the reported gains are under two-fold, which does not support the idea that this approach will have a general role in improving gene editing outcomes.

We agree that the effects of ESR2 inhibition alone are modest at certain loci but can also be strong. Notably, the impact of ESR2 inhibition exceeds that of DNA-PKcs (NHEJ) inhibition alone for the KNL1 target (DNA-PKcs inhibition: 1.3-fold, ESR2 inhibition: 4.3-fold). More important than the sole inhibition of ESR2 for increasing HDR is the synergistic effect when both DNA-PKcs and ESR2 are inhibited. This effect is robust across targets, cell lines, and CRISPR enzymes (increase over DNA-PKcs inhibition alone: 2.2-fold mean, 1.6-fold median; Fig. 3c). We believe that this is an efficient and easily applicable addition to the commonly used DNA-PKcs inhibition in CRISPR editing.

The additional mechanistic data do not fully resolve this. The Western blots show differences that are quite small, generally less than two-fold, and difficult to interpret as strong functional effects.

The mechanistic data obtained from Western blot analysis of cells with and without functional ESR2 are compatible with increased HDR. ESR2-deficient cells treated with the DNA-damaging agent cisplatin exhibit increased levels of the following proteins, which are important for HDR: p-BRCA1 (2.2-fold), BRCA2 (1.3-fold), and ATM (1.5-fold). However, DNA-PKcs is also increased 1.6-fold. In the discussion, we wrote: "The synergistic effect of combining ESR2 inhibition with M3814, could be due to both increased levels of HDR proteins and inhibition of the kinase domain of DNA-PKcs, while the increased levels of the full DNA-PKcs protein may increase the amount of ATM needed for HDR (PMID: 19535303). In the absence of DNA-PKcs protein, levels of ATM, and thus HDR, are reduced (PMID: 18157161)."

Further, we disagree that differences below a two-fold threshold should be dismissed as functionally insignificant in the context of the DNA damage response. The DNA damage checkpoint is a finely tuned signaling cascade in which even modest changes in protein abundance can have profound downstream effects. For instance, we detected increases in ATM protein levels of 1.8- and 1.4-fold, respectively, in ESR2- and AOX1-deficient hiPS cells under basal conditions. ATM is a master regulator of the DNA damage response and has hundreds of known substrates (PMID: 17525332, PMID: 21139141). Given its central role, even relatively small changes in ATM levels can substantially alter the phosphorylation of downstream targets. Furthermore, since enzyme activity is governed by their kinetics, modest increases in ATM protein can amplify checkpoint signaling disproportionately. For the other core DNA repair factors examined in our study, we observed changes ranging from 20% to 220% under DNA damage conditions (Fig. 4d, e). We interpret these changes as biologically meaningful within such a tightly regulated system. We acknowledge that a full mechanistic dissection of ESR2 and AOX1 in DNA repair is an important and exciting question. However, we consider this beyond the scope of the present study.

The immunostaining is even more problematic. Normally, pH2AX and 53BP1 foci correlate tightly, but in Figures 4f and 4g the images suggest an overall increase in 53BP1 staining

without clear punctate foci. This makes the conclusion that ESR2 or AOX1 loss alters repair foci dynamics hard to accept.

We observe punctate 53BP1 foci, but agree that they are difficult to see after compression for PDF conversion. We now upload separate figure files with sufficient resolution (see also below close-up of Fig. 4f). We used ImageJ to count punctate foci per nucleus for the quantification in Fig. 4g.

In Supplementary Figures 15, 16, and 17, we present full microscopic field immunostaining images demonstrating the overlap of 53BP1 and γ H2AX staining following a six-hour treatment with doxycycline. Thanks to your comment, we realized that the γ H2AX labeling was incorrect in these figures, which has now been corrected.

Although 53BP1 recruitment is typically dependent on γ H2AX, it can also occur independently via the MRN (Mre11-Rad50-Nbs1) complex, which is consistent with previous reports (PMID: 38821984, 24094932). In our experiments, we observed clear nuclear localization of 53BP1, indicating its presence at sites of DNA damage. However, we note that technical limitations, including antibody sensitivity, may impact the visibility of punctate γ H2AX foci in immunostaining (staining kit specifications: "Phosphorylation of H2AX can lead to discrete foci or diffuse nuclear staining." Importantly, the elevated levels of ATM observed in our Western blots support active DNA damage signaling, as ATM is the kinase that phosphorylates 53BP1. Thus, our data suggest that ESR2 and AOX1 play a role in modulating DNA repair dynamics.

The drug synergy claims are also not sufficiently supported. In Figure 5, synergy with cisplatin is shown at only a single concentration. Without testing across a range of doses and analyzing the interaction with accepted models of drug synergy, the conclusion remains tentative at best.

In Figure 5d,e, and f we show that the drugs raloxifene, erlotinib, gefitinib, artemether, and olaparib exhibit synthetic lethality in DNA-PKcs-deficient iPSC cells and BRCA2-deficient Capan-1 cells, even without inducing additional cell toxicity by cisplatin. We additionally titrated the cisplatin concentration, selecting one that was toxic but not completely lethal (EC75; total of seven concentrations tested). Similar or even stronger synthetic lethality was observed when cisplatin was combined with the tested drugs in the respective cell lines.

Taken together, while the dataset is large and some of the findings are interesting, the main results remain either context-dependent or incremental. The reliance on high or toxic drug concentrations further limits translational impact. The additional mechanistic experiments help to some degree but do not, in my view, provide strong new functional insights.

We agree that effects can be context-dependent, and we explicitly state the cell types and conditions under which our experiments were performed. Importantly, we mainly used induced pluripotent stem cells which are highly relevant for translational applications.

The point of high or toxic drug concentrations was raised in the previous revision and we believe that we have comprehensively addressed this. We agree that some drugs result in high cell death, but could still be suitable for cell engineering or ex vivo applications. Nevertheless, almost all drugs selected for testing at other targets and cell lines (Fig. 2d, Fig. 5f, Supplementary Fig. 5) are active at concentrations of 2.5 μ M or well below (the only exception being 20 μ M duloxetine, and 10 μ M orphenadrine/tolterodine). One example of a likely non-toxic drug for in vivo use is erlotinib, which results in synthetic lethality in HR- and NHEJ-deficient cells (Fig. 5d, f, Supplementary Fig. 19), because it is effective at similar concentrations as the widely used breast cancer treatment drug olaparib (Fig. 5f). Recognizing the importance of drug toxicity, we utilized our screening data to provide a single toxicity-reduced screening concentration for each drug for future FDA drug repurposing screens (Supplementary Data 2).

Further, our study provide new functional insights as AOX1's role in DNA damage repair and ESR2's influence on genome editing have not been reported previously. While deeper mechanistic exploration than what we have shown here is beyond the scope of this work, we believe that these findings represent meaningful functional advances.

Reviewer #2 (Remarks to the Author):

The authors have responded appropriately to the prior review and the manuscript is now improved. I have no further concerns and am supportive of publication.

Thank you again for the experimental suggestions that improved our manuscript.